# On the Adversarial Risk of Test Time Adaptation: An Investigation into Realistic Test-Time Data Poisoning

**Yongyi Su**[1,4*], **Yushu Li**[1,4*], **Nanqing Liu**[2,4], **Kui Jia**[3], **Xulei Yang**[4], **Chuan-Sheng Foo**[4], **Xun Xu**[4†]

[1]South China University of Technology
[2]Southwest Jiaotong University
[3]The Chinese University of Hong Kong, Shenzhen
[4]Institute for Infocomm Research (I²R), A*STAR

```
{eesuyongyi, eeyushuli}@mail.scut.edu.cn
lansing163@163.com, kuijia@cuhk.edu.cn
{yang_xulei, foo_chuan_sheng, xu_xun}@i2r.a-star.edu.sg
```

## Abstract

Test-time adaptation (TTA) updates the model weights during the inference stage using testing data to enhance generalization. However, this practice exposes TTA to adversarial risks. Existing studies have shown that when TTA is updated with crafted adversarial test samples, also known as test-time poisoned data, the performance on benign samples can deteriorate. Nonetheless, the perceived adversarial risk may be overstated if the poisoned data is generated under overly strong assumptions. In this work, we first review realistic assumptions for test-time data poisoning, including white-box versus grey-box attacks, access to benign data, attack order, and more. We then propose an effective and realistic attack method that better produces poisoned samples without access to benign samples, and derive an effective in-distribution attack objective. We also design two TTA-aware attack objectives. Our benchmarks of existing attack methods reveal that the TTA methods are more robust than previously believed. In addition, we analyze effective defense strategies to help develop adversarially robust TTA methods. The source code is available at `https://github.com/Gorilla-Lab-SCUT/RTTDP`.

## 1 Introduction

Test-time adaptation (TTA) emerges as an effective measure to counter distribution shift at inference stage (Wang et al., 2020; Su et al., 2022; Zhong et al., 2022; Chen et al., 2023; Chi et al., 2024; Wu et al., 2024). Successful TTA methods leverage the testing data samples for self-training (Wang et al., 2020; Su et al., 2024b), distribution alignment (Liu et al., 2021; Su et al., 2022) or prompt tuning (Gao et al., 2022; Liu et al., 2025). Despite the continuing efforts into developing computation efficient and high caliber TTA approaches, the robustness of TTA methods has not picked up until recently, leading to studies examining the robustness of TTA methods under constant distribution shift (Song et al., 2023), correlated testing data stream (Su et al., 2024a; Niu et al., 2023), open-world testing data (Li et al., 2023), adversarial robustness (Wu et al., 2023; Cong et al., 2023), etc. Among these risks, adversarial vulnerability warrants particular attention due to its potential for evading human inspection and the significant consequences of admitting malicious samples during TTA.

Existing research frames the adversarial risk of Test-Time Adaptation (TTA) as the crafting of poisoned testing data, resulting in models updated with such data performing poorly on clean testing samples (Wu et al., 2023; Cong et al., 2023). Consequently, this task is also referred to as Test-Time Data Poisoning (TTDP). The pioneering work DIA (Wu et al., 2023) introduced a poisoning approach by crafting malicious data with access to all benign samples within a minibatch, leveraging real-time model weights for explicit gradient computing, i.e., a white-box attack. Another concurrent

---

*Equal contribution. †Correspondence to <xu_xun@i2r.a-star.edu.sg>. This work was done during Yongyi Su, Yushu Li and Nanqing Liu's visit to I²R.

study (Cong et al., 2023) implements poisoning by preemptively injecting all poisoned data to attack the model even before TTA starts. While these explorations conclude that TTA methods are susceptible to poisoned data, evaluations based on unrealistic assumptions may exaggerate the adversarial risk for several reasons. i) Access to real-time model weights (white-box attack) is often considered overly optimistic, especially given that models are constantly updated during adaptation. Therefore, a grey-box or even black-box attack is preferred for TTDP. ii) The adversary is typically assumed only to be aware of the query samples submitted by themselves. Thus, benign samples submitted by other users should not be utilized for crafting poisoned data, for instance, through bi-level optimization (Wu et al., 2023). iii) Crafting poisoned data requires querying the model with testing samples (Wu et al., 2023). Repeatedly querying the model from a single user could easily trigger alerts in defensive systems. Therefore, any query sample, whether adversarial or benign, should be counted towards the attack budget. iv) Following the above concern, the adversary should not monopolize the entire testing bandwidth. This constraint translates to a scenario where poisoned data only partially occupies the testing stream, and the adversary is not allowed to inject all poisoned data at once even before TTA starts (Cong et al., 2023).

To the best of our knowledge, existing attempts at test-time poisoning have not fully addressed the above realistic concerns. In this work, we aim to propose a threat model that advances towards more Realistic Test-Time Data Poisoning (RTTDP). Firstly, we formulate the threat model under a grey-box attack scenario, where initial model weights are visible. We distill a simple surrogate model from the online model using only the adversary's queries, enabling efficient gradient-based synthesis of poisoned data. Empirical analysis demonstrates that the distilled surrogate provides sufficient information for crafting effective poisoned data. Moreover, to constrain the attack budget, we reformulate the bi-level optimization objective proposed in prior work (Wu et al., 2023) by replacing benign samples with poisoned data only. Through reasoning on generalization error, we illustrate that the attack loss defined on poisoned data can be generalized to benign samples if the distributions between poisoned and benign samples are identical. This insight motivates us to introduce a feature distribution consistency regularization for in-distribution attacks, eliminating the need for additional benign samples to construct the outer objective. Finally, we devise two alternative attack objectives tailored to the unique features of Test-Time Adaptation (TTA) methods. We first propose a high-entropy oriented attack to generate poisoned samples biased towards high entropy. This approach proves effective in compromising TTA methods based on entropy minimization (Wang et al., 2020). However, high-entropy attacks may become less effective with the introduction of simple defense techniques, such as confidence thresholding. Therefore, we explore a low-entropy based attack objective aimed at attacking towards a non-ground-truth class. The combined threat model is applied to a diverse range of TTA methods, resulting in more effective outcomes compared to existing threat models. An overview of the overall framework is presented in Figure 1.

In addition to crafting effective threat models, we delve into exploring practices conducive to enhancing Test-Time Adaptation (TTA)'s adversarial robustness. Contrary to the reliance on adversarially trained models (Madry et al., 2018) and robust batch normalization estimation (Wu et al., 2023), we draw inspiration from empirical observations of robust TTA methods. Our validation reveals that confidence thresholding, data augmentation, exponential moving averaging (EMA), and random parameter restoration represent potential directions for improving the adversarial robustness of TTA methods.

We summarize the contributions of this work as follows.

- We argue the unrealistic assumptions, e.g. white-box attack, access to benign subset and offline attack order, made in existing attempts at TTDP may overestimate the adversarial risk of TTA methods. To address this, we first propose key criteria for defining realistic test-time data poisoning scenarios.

- Under our proposed realistic test-time protocol, we analyze the generalization error and introduce an in-distribution attack strategy with feature consistency regularization. This strategy eliminates the need for additional benign samples in evaluating the outer objective. In addition, we tailor attack objectives specifically for TTA methods, resulting in more effective poisoning.

- We perform extensive evaluations on state-of-the-art TTA methods, demonstrating the efficacy of our proposed in-distribution attack strategy. Furthermore, we identify certain practices that are conducive to improving realistic adversarial robustness.

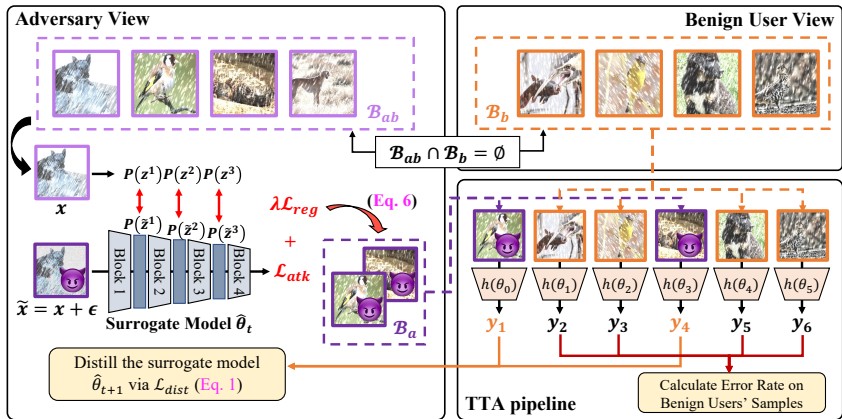

Figure 1: Illustration of the proposed Realistic Test-Time Data Poisoning (RTTDP) pipeline. $\mathcal{B}_{ab}$ indicates the adversary benign subset, and $\mathcal{B}_a$ indicates the adversary poisoned subset where the samples are poisoned from the clean samples in $\mathcal{B}_{ab}$. $\mathcal{B}_b$ indicates the benign users' subset where the samples are used to validate the adversarial risk of TTA pipeline and these samples cannot be access by the adversary. Adversary generates poisoned data by attacking a regularized objective without accessing benign samples from other users. Model is attacked when carrying out TTA on testing data stream mixed with benign and poisoned data.

## 2 RELATED WORK

**Test-Time Adaptation (TTA)**: Wang et al. (2020; 2022); Niu et al. (2022); Su et al. (2022); Li et al. (2023); Liang et al. (2024); Wu et al. (2024) and Wang et al. (2025) have shown significant success in bridging the domain gap by using stream-based testing samples to dynamically update models in real time. The success of TTA is mainly attributed to the self-supervised learning on testing data. While it has proved sensitive to confirmation bias (Arazo et al., 2020), many solutions were proposed to minimize the influences of the wrong pseudo-labels, including minimizing sample entropy (Wang et al., 2020; Liang et al., 2020), adding regularization terms (Song et al., 2023; Su et al., 2024b), using confidence thresholding (Niu et al., 2022; 2023), updating models with exponential moving average architectures (Wang et al., 2022; Döbler et al., 2023), partially updating model weights (Wang et al., 2020; Yuan et al., 2023), and augmenting testing samples (Zhang et al., 2022; Döbler et al., 2023), all aimed at minimizing sample distribution discrepancies. However, adaptation during the testing stage remains highly risky. In this work, we aim to investigate the risk of TTA posed by data poisoning.

**Robustness in Test-Time Adaptation**: Recent research has increasingly focused on the robustness of TTA in realistic deployments. Wang et al. (2022) and Brahma & Rai (2023) tackle issues of catastrophic forgetting due to changing test data distributions. Niu et al. (2023), Gong et al. (2022) and Yuan et al. (2023) address non-i.i.d. and shifting label distributions in test data. Li et al. (2023) and Zhou et al. (2023) introduce open-world scenarios in TTA, where test data may include novel classes not present in the source domain. Furthermore, Wu et al. (2023) and Cong et al. (2023) investigate the threat of data poisoning in TTA, where attackers alter test data to exploit vulnerabilities in TTA methods. DIA (Wu et al., 2023) uses bi-level optimization to degrade target sample performance, assuming white-box attack access. In contrast, TePA (Cong et al., 2023) includes an exclusive offline stage for poisoning data on the source model prior to the TTA process. Inspired by these works, we focus on the severe threat of data poisoning. We examine the adversarial robustness of TTA under realistic conditions, with access only to the source model and partial test samples for data poisoning. Our attack, despite these constraints, outperforms previous methods, highlighting the significant adversarial risks that TTA methods face.

**Adversarial Attack & Data Poisoning**: Adversarial risk is a crucial concern for models to address for safe deployment, which can be divided into two categories: adversarial attacks and data poisoning. Adversarial attacks (Szegedy et al., 2014; Akhtar & Mian, 2018; Goodfellow et al., 2014; Madry et al., 2018; Croce & Hein, 2020; Chen et al., 2022; Chakraborty et al., 2018) manipulate models during inference by adding small perturbations to input data. White-box attacks (Szegedy et al., 2014; Tramèr et al., 2018) assume full access to the victim model, creating adversarial examples by maximizing loss gradients. In contrast, grey-box attacks (Chen et al., 2017; Ilyas et al., 2018; Ru et al., 2019) assume no model access, generating adversarial samples by estimating gradients through intensive querying. Data poisoning (Biggio et al., 2012; Yang et al., 2017; Shafahi et al., 2018; Alfeld et al., 2016; Huang et al., 2021; Fowl et al., 2021; Fan et al., 2022) compromises models by injecting

manipulated data into the training set, misleading the training process. Traditional data poisoning assumes that the attacker can observe and poison the entire training set at once. Recently, online poisoning (Zhang et al., 2020) relaxes this assumption by requiring knowledge of the model updating strategies. In this work, we explore a more realistic scenario of TTA's adversarial robustness, focusing on data poisoning without online model access and without multiple queries for the same data.

## 3 SETTING: REALISTIC TEST-TIME DATA POISONING

### 3.1 OVERVIEW OF TEST-TIME DATA POISONING

We first provide a generic overview of test-time adaptation for a $K$-way classification task. We denote the testing data as $\mathcal{D} = \{x_i\}_{i=1}^{N_t}$ and a pre-trained model as $\theta_0$. TTA methods often employ unsupervised loss, $\mathcal{L}_{tta}$, to update model parameters upon observing a minibatch of testing samples $\mathcal{B}_t = \{x_i\}_{i=1}^{N_b}$ at timestamp $t$. Usually, a subset of model parameters $\theta_t^u \subseteq \theta_t$ is subject to update at timestamp $t$, and we denote the BN statistics as $\theta^b(\mathcal{B}_t) = \{\mu(\mathcal{B}_t), \sigma^2(\mathcal{B}_t)\}$ and the frozen parameters as $\theta^f$. The posterior of the sample $x_i$ towards the model parameter $\theta$ is $h(x_i; \theta) \in [0,1]^K$. The adversarial risk arises when a subset of testing samples are poisoned, e.g. through adding an adversarial noise $\tilde{x} = x + \epsilon$, $s.t.$ $||\epsilon||_\infty \leq b$. The model trained on poisoned data exhibit poor performance on clean/benign testing samples. In a typical TTA scenario, the online model is queried by both adversary and benign users. Thus, we denote the query data from adversary as **adversary poisoned subset** $\mathcal{B}_a = \{\tilde{x}_i\}$. The poisoned subset could be generated from arbitrary clean testing data, denoted as $\mathcal{B}_{ab} = \{x_i\}$ (e.g. use any public clean images). The generation follows an additive noise, i.e. $\tilde{x}_i = x_i + \epsilon_i$, $s.t.$ $\tilde{x}_i \in \mathcal{B}_a$ $x_i \in \mathcal{B}_{ab}$, where the noise $\epsilon_i$ is the data poisoning to be learned. The query data from benign users is denoted as **benign subset** $\mathcal{B}_b = \{x_i\}$, where $\mathcal{B}_{ab} \cap \mathcal{B}_b = \emptyset$. Generally, we form the combination of both, $\mathcal{B}_t = \mathcal{B}_a \cup \mathcal{B}_b$, as a single TTA minibatch. The effectiveness of test-time poisoning is evaluated at the attack success rate (classification error) on benign subset $\mathcal{B}_b$. An illustration of batch split is presented in the Appendix A.1.1.

### 3.2 REALISTIC TEST-TIME DATA POISONING PROTOCOL

The adversarial risk of TTA methods must be assessed under realistic attacks. Existing works may have exaggerated the adversarial risk when attack is implemented under an overly strong assumption. We summarize the criteria that define a more realistic attack as follows.

**White-Box v.s. Grey-Box Attack:** Access to real-time TTA model parameters is a key factor for crafting realistic test-time data poisoning. Contrary to the assumption adopted by white-box adversarial attack (Szegedy et al., 2014; Biggio et al., 2013) that a frozen model is deployed for inference, the TTA model parameters experience constant updating at inference stage. The update is performed on the cloud side, thus the attacker doesn't normally have access to real-time model weights. Such a realistic assumption prompts us to explore a more relaxed grey-box test-time data poisoning, i.e. only the model architecture and initial model weights are available to the adversary, such as the open-source famous pre-trained models (He et al., 2016; Dosovitskiy et al., 2021) and popular foundation models (Radford et al., 2021; Kirillov et al., 2023; Oquab et al., 2024).

**Access To Benign Subset:** The effectiveness of TTDP is evaluated on the benign subset $\mathcal{B}_b$ submitted by benign users. In a standard cloud service, users are generally restricted from accessing the queries of other users. Thus the adversary should only have access to adversary subset $\mathcal{B}_a$. This assumption prohibits the practice of crafting poisoned data by directly optimizing (minimizing) the loss on benign subset (Wu et al., 2023), on which the attack success rate (performance) is calculated.

**Attack Order:** Finally, attacking TTA model in a realistic way should be implemented during the adaptation stage. Attacking the model before TTA begins is deemed less practical (Cong et al., 2023).

Based on the aforementioned key criteria, we provide a summary of existing test-time data poisoning methods in Tab. 1. Our analysis indicates that none of the current methods fully satisfy all the established criteria. Specifically, DIA (Wu et al., 2023) employs a white-box attack strategy, generating poisoned samples by maximizing the error rate on a subset of benign data. TePA (Cong et al., 2023) attacks the pre-trained model with an offline surrogate model prior to the commencement of test-time adaptation. In the rest of the paper, we stick to the most realistic assumptions, i.e. online grey-box poisoning and no access to benign subset, named as **Realistic Test-Time Data Poisoning (RTTDP)**.

Table 1: Taxonomy of methods based on the criteria for realistic test-time data poisoning.

| Setting | Grey-box v.s. White-box | Access to Benign Subset | Attack Order |
|---|---|---|---|
| DIA (Wu et al., 2023) | White-box | ✓ | Online |
| TePA (Cong et al., 2023) | Grey-box | ✗ | Offline |
| **RTTDP (Ours)** | Grey-box | ✗ | Online |

**Adversarial Attacks on Standard Image Classification**: Common grey-box or black-box adversarial attack techniques are often impractical for RTTDP due to two primary challenges. First, existing adversarial attack methods **operate on a static model and inherently require multiple queries** for gradient approximation or fitness evaluation, as seen in query-based attacks (Li et al., 2020; Xu et al., 2021), genetic algorithms (Chen et al., 2019), and black-box optimization (Qiu et al., 2021). However, in the test-time adaptation setting, the online model is continuously updated with each query during the inference phase. Thus, repetitive querying the model for gradient approximation or fitness evaluation is unavailable. Second, traditional adversarial attack methods focus on crafting adversarial samples to degrade the performance of the attacker's own input data. In contrast, in a realistic test-time data poisoning scenario, poisoned samples are introduced into the test-time adaptation process to **degrade the performance of benign samples submitted by other users**.

## 4 METHODOLOGY

### 4.1 GREY-BOX ATTACK BY SURROGATE MODEL DISTILLATION

To tackle the challenge of access to TTA model parameters, we propose to maintain a surrogate model, denoted as $\hat{\theta}_t$ at timestamp $t$, for the purpose of synthesizing poisoned data. To ensure good approximation, we distill the target model $\theta_t$ into the surrogate model $\hat{\theta}_t$ by leveraging the feedback of poisoned data from the online target model. Specifically, for each query to the target model, we minimize the symmetric KL-Divergence between the posteriors of the target and surrogate models, as Eq. 1. Our empirical observations demonstrate that utilizing the adversarial subset $\mathcal{B}_a$ for distillation yields performance comparable to a white-box attack, as illustrated in Fig. 2 (a).

$$\mathcal{L}_{dist} = \frac{1}{|\mathcal{B}_a|} \sum_{x_i \in \mathcal{B}_a} \frac{1}{2} \left[ KLD\left( h(x_i; \theta_t) || h(x_i; \hat{\theta}_t) \right) + KLD\left( h(x_i; \hat{\theta}_t || h(x_i; \theta_t)) \right) \right] \quad (1)$$

### 4.2 IN-DISTRIBUTION TEST-TIME DATA POISONING

In this section, we further address the challenge of attacking TTA model without access to benign user's testing samples. In the first place, we revisit DIA (Wu et al., 2023), which formulated test-time poisoned data generation as a bi-level optimization problem, in Eq. 2.

$$\min_{\mathcal{B}_a} \frac{1}{|\mathcal{B}_b|} \sum_{x_i \in \mathcal{B}_b} \mathcal{L}_{atk}\left( x_i; \theta_t^*(\mathcal{B}_t) \right)$$
$$s.t. \ \mathcal{B}_t = \mathcal{B}_a \cup \mathcal{B}_b; \ \theta^{b\prime} = \{\mu(\mathcal{B}_t), \sigma^2(\mathcal{B}_t)\}; \quad (2)$$
$$\theta_t^{u*} = \arg\min_{\theta_t^u} \mathcal{L}_{tta}(\mathcal{B}_t; \theta_t^*(\mathcal{B}_t)); \ \theta_t^*(\mathcal{B}_t) = \theta_t^{u*} \cup \theta^{b\prime} \cup \theta^f$$

The above bi-level optimization problem employed in DIA (Wu et al., 2023) aims to generate adversarially poisoned data $\mathcal{B}_a$ by minimizing the loss function $\mathcal{L}_{atk}$, which is computed on the benign samples $\mathcal{B}_b$. Although DIA approximates $\theta_t^{u*} \approx \theta_t^u$ to discard the inner TTA gradient update loop and reduce the number of queries to the online model, several realistic concerns still persist under the RTTDP protocol. First, as discussed in Sec. 3.2, DIA, as a white-box attack method, must query the online TTA model $\theta_t^*$ for generating poisoned samples. To mitigate this issue, we propose to leverage a surrogate model $\hat{\theta}_t$ as an proxy model, which is distilled by Eq. 1 with the last feedback of $\mathcal{B}_{a,t-\delta}$, where $\delta$ denotes the time interval between two injected poisoned subsets. Second, DIA evaluates the outer optimization by employing the benign samples $\mathcal{B}_b$ (assuming access to benign users' query samples). The adversarial risk mainly arises from the injection of poisoned samples into the TTA training process. However, employing the validation (benign) samples as the outer optimization objective may result in an overestimation of this risk, as the specific poisoned samples could be tailored for the attack on the inference of the current batch of benign samples (Park et al., 2024), e.g., $\theta^{b\prime}$, rather than for attacking the TTA process, i.e., $\theta^u$.

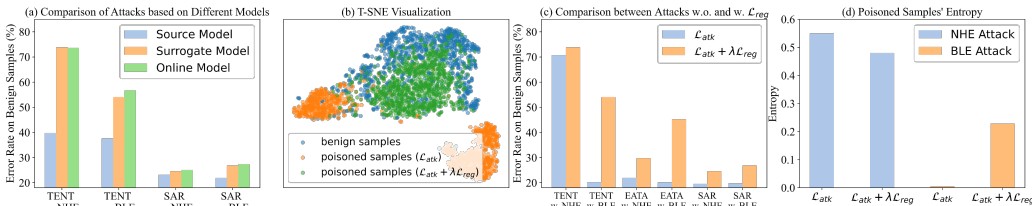

Figure 2: (a) The attack performance comparison about the poisoned samples generated on Source (pretrained) Model, our proposed Surrogate Model and Online target Model (white box). (b) The T-SNE visualization of the feature points (before FC layer). Without $\mathcal{L}_{reg}$, common attack losses (e.g. maximizing cross-entropy) produce poisoned samples (orange dots) that are far from benign ones (blue dots), leading to less effective attacks. (c) The attack performance comparison between w.o. and w. $\mathcal{L}_{reg}$. (d) The average prediction entropy of the poisoned samples generated by our proposed two different attack objectives, respectively.

To prevent from using benign users' samples for optimization and assuming grey-box attack, one possible solution is to swap the benign users' samples $\mathcal{B}_b$ with the adversary's clean sample $\mathcal{B}_{ab}$ based on the assumption that $\mathcal{B}_{ab}$ and $\mathcal{B}_b$ are drawn from similar distributions, and replacing the white-box model $\theta$ with distilled model $\hat{\theta}$ and discarding the inner TTA loop $\hat{\theta}_t^* \approx \hat{\theta}_t$ as adopted by DIA (Wu et al., 2023), resulting in the following formulation.

$$\min_{\mathcal{B}_a} \frac{1}{|\mathcal{B}_{ab}|} \sum_{x_i \in \mathcal{B}_{ab}} \mathcal{L}_{atk}\left(x_i; \hat{\theta}_t'(\mathcal{B}_t)\right) \tag{3}$$

$$s.t. \quad \mathcal{B}_t = \mathcal{B}_a \cup \mathcal{B}_{ab}; \quad \hat{\theta}^{b'} = \{\mu(\mathcal{B}_t), \sigma^2(\mathcal{B}_t)\}; \quad \hat{\theta}_t'(\mathcal{B}_t) = \hat{\theta}_t^u \cup \hat{\theta}^{b'} \cup \hat{\theta}^f$$

Despite the above formulation alleviates the assumption of the available to access the benign users' samples and the online model parameters, it still remains elusive to tackle. First, as most TTA methods leverage the current batch statistics to estimate the statistics on target domain, forwarding $\mathcal{B}_t$ results in esimating the BN statistics on the combination of $\mathcal{B}_a$ and $\mathcal{B}_{ab}$ which bias update of BN parameters. This may results in a mismatch between the feature distribution of $\mathcal{B}_{ab}$ and $\mathcal{B}_b$, if $\mathcal{B}_b$ is queried independently. Thus, the poisoning effect may fail to generalize to $\mathcal{B}_b$. This is evidenced by an empirical study into the distribution of poisoned data in Fig. 2 (b) where the distribution of generated poisoned data (orange dots) deviate substantially from the benign samples (blue dots) if the Eq. 3 is directly attacked. Therefore, we are prompted to explore a solution that does not explicitly require forward pass for both $\mathcal{B}_a$ and $\mathcal{B}_{ab}$ simultaneously and is able to transfer the attacked effect from $\mathcal{B}_a$ to the benign subset i.e. $\mathcal{B}_{ab}$ or $\mathcal{B}_b$.

To address this challenge, we propose introducing additional constraints and integrating the optimized target $\mathcal{B}_{ab}$ with the optimizing objective $\mathcal{B}_a$ into a unified objective $\mathcal{B}_a$, as presented in Eq. 4, where $D(P_1, P_2)$ is a metric of two distributions and $P_a$ and $P_{ab}$ refer to the feature distributions of $\mathcal{B}_a$ and $\mathcal{B}_{ab}$. We have the follow reason why the formulation is effective. If $\mathcal{B}_a$ and $\mathcal{B}_{ab}$ have the similar distribution, we can have $P_a \rightarrow P_{ab} \Rightarrow \mathbb{E}_{P_a(x)}[\mathcal{L}_{atk}(x)] \rightarrow \mathbb{E}_{P_{ab}(x)}[\mathcal{L}_{atk}(x)]$ according to the Probably Approximately Correct (PAC) learning framework (Valiant). Thus attack against $\mathcal{B}_a$ has a high chance to generalize to $\mathcal{B}_{ab}$.

$$\min_{\mathcal{B}_a} \frac{1}{|\mathcal{B}_a|} \sum_{x_i \in \mathcal{B}_a} \mathcal{L}_{atk}\left(x_i; \hat{\theta}_t'(\mathcal{B}_a)\right); \quad s.t. \quad D(P_a, P_{ab}) = 0 \tag{4}$$

**Feature Consistency Regularization for In-Distribution Attack**: To achieve indistinguishable distribution between $P_a$ and $P_{ab}$, we propose to measure the discrepancy at feature level and introduce the discrepancy as a constraint to the optimization problem. Crucially, since the attack loss is defined in the representation extracted by the backbone network, the distribution consistency is ideally imposed on the intermediate features except for the final semantic one. Specifically, we denote the $l$-th intermediate feature map before each normalization layer as $z_i^l = f^l(x_i) \in \mathbb{R}^{H_l \times W_l \times D_l}$. A single Gaussian distribution is fitted to intermediate layer features, as $\mu_i^l = \frac{1}{H_l W_l} \sum_{h,w} z_{ihw}^l$, $\Sigma_i^l = \frac{1}{H_l W_l} \sum_{h,w} (z_{ihw}^l - \mu_i^l)(z_{ihw}^l - \mu_i^l)^\top$ and $\tilde{\mu}_i^l = \frac{1}{H_l W_l} \sum_{h,w} \tilde{z}_{ihw}^l$, $\tilde{\Sigma}_i^l = \frac{1}{H_l W_l} \sum_{h,w} (\tilde{z}_{ihw}^l - \tilde{\mu}_i^l)(\tilde{z}_{ihw}^l - \tilde{\mu}_i^l)^\top$, where $\tilde{z}_i$ and $z_i$ refer to the sample features from $\mathcal{B}_a$ and $\mathcal{B}_{ab}$, respectively. The KL-Divergence between feature distributions is introduced as the constraint.

$$\mathcal{L}_{reg} = \frac{1}{L} \sum_l KLD(\mathcal{N}(\mu_i^l, \Sigma_i^l) || \mathcal{N}(\tilde{\mu}_i^l, \tilde{\Sigma}_i^l)) \tag{5}$$

With the introduced constraint $\mathcal{L}_{reg} = 0$, we finally formulate the problem as Eq. 6. The problem now degenerates to a single level optimization with constraints which can be easily converted into an unconstrained optimization problem via Lagrangian multiplier (Lag, 2008). The unconstrained problem can be solved in an iterative fashion with detailed algorithm presented in the Appendix A.1.6.

$$
\begin{aligned}
&\min_{\mathcal{B}_a} \frac{1}{|\mathcal{B}_a|} \sum_{x_i \in \mathcal{B}_a} \mathcal{L}_{atk}\left(x_i; \hat{\theta}'_t(\mathcal{B}_a)\right) \\
&s.t. \quad \hat{\theta}^{b\prime} = \{\mu(\mathcal{B}_a), \sigma^2(\mathcal{B}_a)\}; \quad \hat{\theta}'_t(\mathcal{B}_a) = \hat{\theta}^u_t \cup \hat{\theta}^{b\prime} \cup \hat{\theta}^f; \quad \mathcal{L}_{reg} = 0 \\
\Rightarrow &\min_{\mathcal{B}_a} \max_{\lambda} \frac{1}{|\mathcal{B}_a|} \sum_{x_i \in \mathcal{B}_a} \mathcal{L}_{atk}\left(x_i; \hat{\theta}'_t(\mathcal{B}_a)\right) + \lambda \mathcal{L}_{reg} \\
\Rightarrow &\min_{\mathcal{B}_a} \max_{\{\lambda_0 \cdots \lambda_{L-1}\}} \frac{1}{|\mathcal{B}_a|} \sum_{x_i \in \mathcal{B}_a} \mathcal{L}_{atk}\left(x_i; \hat{\theta}'_t(\mathcal{B}_a)\right) + \frac{1}{L} \sum_l \lambda_l KLD(\mathcal{N}(\mu^l_i, \Sigma^l_i) || \mathcal{N}(\tilde{\mu}^l_i, \tilde{\Sigma}^l_i)),
\end{aligned}
\tag{6}
$$

where $L$ is the number of intermediate feature layers. Through optimizing the above objective, now we could craft the effective poisoned data $\mathcal{B}_a$ that satisfies $\mathbb{E}_{x_i \in \mathcal{B}_a} \mathcal{L}_{atk}(x_i; \hat{\theta}_t) \approx \mathbb{E}_{x_i \in \mathcal{B}_b} \mathcal{L}_{atk}(x_i; \hat{\theta}_t)$, since the approximation equation $P_a \to P_{ab} \to P_b$ now holds. Fig. 2 (c) demonstrates the effectiveness of this in-distribution attack, where the attack performance of the attack objective combined with the feature regularization (orange bars) is always higher than that of the corresponding attack objective alone (blue bars) in different TTA methods. Next, we would introduce the attack objectives $\mathcal{L}_{atk}$ that we designed to effectively generate different kinds of poisoned samples.

## 4.3 TTA-Aware Attack Objective

The specific design of attack objective $\mathcal{L}_{atk}$ warrants careful consideration. The existing works examined both targeted and indiscriminative attacks, demonstrating that both are effective against state-of-the-art TTA methods (Wu et al., 2023). However, we argue that an attack objective is not universally effective against all TTA methods. Therefore, we investigate two types of attack objectives as follows, and the prediction entropy of poisoned samples generated via the proposed attack losses can be compared in Fig. 2 (d).

**High Entropy Attack Objective**: Self-Training based TTA methods, e.g. TENT, RPL, are vulnerable for out-of-distribution samples with high entropy (Niu et al., 2023), and therefore, a straightforward way to generate poisoned data to attack TTA model is by maximizing the entropy of poisoned data as proposed in TePA (Cong et al., 2023) since these poisoned samples with high prediction entropy would induce high updating gradient. However, maximizing the prediction entropy does not guarantee wrong pseudo labels. Therefore, we propose a stronger high entropy attack objective called Notch High Entropy Attack (**NHE Attack**). Based on the uniform distribution, we set the probability in the ground-truth label to zero and construct the target distribution $Q$. Then we minimize the cross-entropy against target distribution $Q$.

$$
\mathcal{L}^{NHE}_{atk}(\tilde{x}_i) = -\sum_k Q_{ik} \log h_k(\tilde{x}_i) \quad s.t. \quad Q_{ik} = \begin{cases} 0 & k = y_i \\ \frac{1}{K-1} & others \end{cases}
\tag{7}
$$

**Low Entropy Attack Objective**: High entropy attack objective is particular effective against self-training based TTA methods because of high updating gradient, yet they could be easily defended by some defense strategies, e.g. entropy thresholding. Therefore, we further explore a new low entropy based attack objective. DIA (Wu et al., 2023) proposed to maximize the cross-entropy loss on the benign samples (indiscriminate attack) to generate the other poisoned samples. However, we empirically found that maximizing the cross-entropy loss without any constraints on one sample is prone to maximizing the probability of the most confident class (except the ground-truth) of one model, and feeding these samples into TTA model would bring up the following issues. i) The model will quickly bias towards the most confident class and collapse if without any class diversity constraints. ii) If class diversity constraints are applied (assembled in several TTA methods, e.g. EATA, ROID), this objective will become less ineffective, since class-biased poisoned samples will obtain a less updating weighting than other benign samples. Therefore, we propose a class-balanced low entropy attack, termed Balanced Low Entropy Attack (**BLE Attack**). Specifically, we maintain an moving average probability confusion $C \in [0,1]^{K \times K}$ to store the prediction bias in each class and find a global optimal label mapping $M \in \{0,1\}^{K \times K}$ such that each class is attacked towards the most probable non ground-truth class. Details of deriving label mapping $M$ is deferred to the

Appendix A.1.2. Finally, the BLE Attack objective is calculated as Eq. 8.

$$\mathcal{L}_{atk}^{BLE}(\tilde{x}_i) = -\sum_k \mathbb{1}(k = \arg\max_{q \neq y_i} M_{y_i,q}) \log h(\tilde{x}_i) \qquad (8)$$

**Overall Attack Strategy**: We craft poisoned data by attacking the aforementioned in-distribution attack objective with regularization. Following the practice that poisoned data should be less discernible by human, we employ a 40 steps Projected Gradient Descent algorithm (Boyd & Vandenberghe, 2004) on the combined objective in Eq. 6 with a budget $b$. More details of the whole data poisoning algorithm are deferred to the Appendix A.1.6.

## 5 EXPERIMENT

### 5.1 EXPERIMENT DETAILS

**Benchmark Poisoning Methods**: We evaluated the following methods under our proposed RTTDP setting. **Unlearnable Examples** (Huang et al., 2021) generates the poisoned noise by minimizing the cross-entropy of $\mathcal{B}_a$ on a randomly initialized model. **Adversarial Poisoning** (Fowl et al., 2021) proposed to minimize the cross-entropy between the posterior probabilities of poisoned samples, $\mathcal{B}_a$, and their corresponding incorrect labels, $\hat{y}$, where the incorrect labels are defined as $\hat{y}_i = y_i + 1$. **DIA** (Wu et al., 2023) is one of the first approaches towards test-time data poisoning, generating the poisoned data via maximizing the cross-entropy of other benign data. We adapt DIA to the realistic evaluation protocol by splitting $\mathcal{B}_{ab}$ into two subsets of 50% each, i.e. $\mathcal{B}_{ab}^p : \mathcal{B}_{ab}^b = 1 : 1$, and craft $\mathcal{B}_a^p$ to maximize the cross-entropy of $\mathcal{B}_{ab}^b$. **TePA** (Cong et al., 2023) proposed to maximize entropy to generate poisoned data and performed attack before TTA starts. We adapt TePA to generate the poisoned data based on the source model and inject them into TTA pipeline on-the-fly. **MaxCE** (Madry et al., 2018) is an established way to create adversarial samples by maximizing the cross-entropy loss. Finally, we evaluate the two attack objectives proposed in this paper, i.e high entropy attack (**NHE Attack**) and low entropy attack against most probable and balanced non ground-truth class (**BLE Attack**). For both NHE Attack and BLE Attack, we evaluate the attack objective subject to the constraint of our proposed feature consistency (Eq. 6). For all methods that require gradient-based optimization, we employ the Projected Gradient Descent (PGD) algorithm (Boyd & Vandenberghe, 2004) to perform the constrained optimization. We use 40 steps PGD for all methods for a fair comparison.

**Datasets**: We evaluate on three datasets, widely adopted for TTA benchmarking. **CIFAR10-C**, **CIFAR100-C** and **ImageNet-C** are synthesized from the original clean validation set by adding various types of corruptions to simulate natural distribution shifts (Hendrycks & Dietterich, 2019). We choose corruption level 5 and perform continual test-time adaptation setting (Wang et al., 2022) for evaluation. Following prior works (Wang et al., 2022; Döbler et al., 2023), we adopt the pre-trained WideResNet-28 (Zagoruyko & Komodakis, 2016), ResNeXt-29 (Xie et al., 2017), and ResNet-50 (He et al., 2016) models for experiments on the CIFAR10-C, CIFAR100-C, and ImageNet-C datasets, respectively. More experiment details can be found in the Appendix A.3.

### 5.2 EVALUATION ON TEST-TIME DATA POISONING

We present the results of comparing different attack objectives against state-of-the-art TTA methods in Tab. 2, Tab. 3 and Tab. 4 for CIFAR10-C, CIFAR100-C and ImageNet-C respectively. We make the following observations from the results. i) Contrary to the claims that TTA methods are extremely vulnerable to data poisoning, under the realistic data poisoning protocol, **without accessing to benign data, it's not trivial to transfer the adversarial risk from poisoned data to benign data,** especially for the TTA methods using EMA model such as CoTTA and ROID. In particular, existing methods do not pose too much risk to more advanced TTA methods without feature consistency regularization. DIA and TePA are more effective on TENT and RPL than other TTA methods. We attribute this to the fact that both TENT and RPL are naive self-training methods without filtering testing samples, hence, poisoned data could easily mislead model update. ii) Our proposed two attack objectives generally perform better than existing poisoning methods, demonstrating a better average ranking and a higher average error rate. This is attributed to the combination of the well-designed attack objective and the regularization of feature consistency. On the other hand, low entropy attack (BLE) obtains significantly improved with our proposed feature consistency compared with the similar low entropy attack i.e. MaxCE. iii) "Non-uniform" attack in general yields higher attack success rate

Table 2: Evaluation of test-time data poisoning under the RTTDP protocol for CIFAR10-C. We report the attack success rate (higher the better) for each TTA method and the average ranking (lower the better) for each attack objective. * indicates that the method is modified to align with the RTTDP protocol.

| Attack Freq. | Attack Objective | Source | TENT | RPL | EATA | TTAC | SAR | CoTTA | ROID | Avg. Err. (↑) | Avg. Rank (↓) |
|---|---|---|---|---|---|---|---|---|---|---|---|
| Uniform | No Attack | | 19.72 | 21.00 | 18.03 | 17.41 | 18.94 | 16.46 | 16.37 | 18.28 | 7.43 |
| | Unlearnable Examples (Huang et al., 2021) | | 32.61 | 26.62 | 20.11 | 18.43 | 19.23 | 17.27 | 17.80 | 21.72 | 4.86 |
| | Adversarial Poisoning (Fowl et al., 2021) | | 19.60 | 19.90 | 18.94 | 18.69 | 19.90 | **18.34** | **19.12** | 19.21 | 3.86 |
| | DIA* (Wu et al., 2023) | 43.81 | 26.04 | 21.87 | 18.94 | 18.56 | 19.46 | 17.72 | 17.77 | 20.05 | 4.86 |
| | TePA* (Cong et al., 2023) | | 33.78 | 22.36 | 23.37 | 17.75 | 19.53 | 16.57 | 18.76 | 21.73 | 4.43 |
| | MaxCE (Madry et al., 2018) | | 18.55 | 20.81 | 18.17 | 18.50 | 19.50 | 16.88 | 18.57 | 18.71 | 6.00 |
| | BLE Attack (Ours) | | 54.07 | 51.99 | **45.20** | **34.00** | **26.80** | 18.12 | 19.06 | 35.61 | **1.57** |
| | NHE Attack (Ours) | | **73.86** | **72.40** | 29.73 | 18.67 | 24.56 | 17.54 | 17.00 | **36.25** | 2.86 |
| Non-Uniform | No Attack | | 19.29 | 20.36 | 17.75 | 16.89 | 18.74 | 16.18 | 15.81 | 17.86 | 5.71 |
| | DIA* (Wu et al., 2023) | | 22.84 | 25.70 | 19.75 | 18.35 | 19.42 | 19.18 | 17.79 | 20.43 | 4.00 |
| | TePA* (Cong et al., 2023) | 43.55 | 40.31 | 32.32 | 23.03 | 18.06 | 19.49 | 18.07 | 18.50 | 24.25 | 3.43 |
| | MaxCE (Madry et al., 2018) | | 18.64 | 20.01 | 18.29 | 18.43 | 19.47 | 18.01 | 18.94 | 18.83 | 4.43 |
| | BLE Attack (Ours) | | 56.17 | 46.66 | **50.97** | **34.25** | **27.54** | 19.23 | **20.12** | 36.42 | **1.43** |
| | NHE Attack (Ours) | | **74.93** | **73.65** | 27.56 | 18.75 | 24.95 | **20.86** | 16.77 | **36.78** | 2.00 |

Table 3: Evaluation of test-time data poisoning under the RTTDP protocol for CIFAR100-C. We report the attack success rate (higher the better) for each TTA method and the average ranking (lower the better) for each attack objective. * indicates that the method is modified to align with the RTTDP protocol.

| Attack Freq. | Attack Objective | Source | TENT | RPL | EATA | TTAC | SAR | CoTTA | ROID | Avg. Err. (↑) | Avg. Rank (↓) |
|---|---|---|---|---|---|---|---|---|---|---|---|
| Uniform | No Attack | | 60.25 | 47.12 | 32.20 | 31.93 | 31.62 | 32.13 | 29.11 | 37.77 | 7.29 |
| | Unlearnable Examples (Huang et al., 2021) | | 75.31 | 72.74 | 37.59 | 33.66 | 41.75 | 32.66 | 31.31 | 46.43 | 3.71 |
| | Adversarial Poisoning (Fowl et al., 2021) | | 33.87 | 34.27 | 32.11 | 35.49 | 32.27 | 32.76 | 30.15 | 32.99 | 5.71 |
| | DIA* (Wu et al., 2023) | 46.23 | 74.79 | 68.42 | 33.77 | 32.57 | 33.20 | 32.68 | 30.10 | 43.65 | 5.29 |
| | TePA* (Cong et al., 2023) | | 75.79 | 81.98 | 36.74 | 33.41 | 35.79 | 32.41 | 32.24 | 46.91 | 3.57 |
| | MaxCE (Madry et al., 2018) | | 42.36 | 39.04 | 34.08 | **40.73** | 32.01 | 32.18 | 31.08 | 35.93 | 5.57 |
| | BLE Attack (Ours) | | 73.93 | 76.71 | **47.30** | 34.58 | 43.25 | **32.81** | **32.27** | 48.69 | **2.28** |
| | NHE Attack (Ours) | | **92.08** | **91.72** | 37.86 | 33.85 | **56.09** | 32.50 | 31.48 | **53.65** | 2.57 |
| Non-Uniform | No Attack | | 62.30 | 49.69 | 31.45 | 32.07 | 31.37 | 32.56 | 28.39 | 38.26 | 5.71 |
| | DIA* (Wu et al., 2023) | | 76.83 | 71.44 | 33.74 | 32.63 | 33.24 | 33.13 | 30.07 | 44.44 | 4.14 |
| | TePA* (Cong et al., 2023) | 46.33 | 82.29 | 88.48 | 36.40 | 34.89 | 35.56 | 33.44 | 33.20 | 49.19 | 2.43 |
| | MaxCE (Madry et al., 2018) | | 41.89 | 38.05 | 33.20 | **42.12** | 31.90 | 32.78 | 31.66 | 35.94 | 4.43 |
| | BLE Attack (Ours) | | 71.30 | 72.97 | **47.21** | 35.26 | 41.22 | 33.50 | 32.60 | 47.72 | 2.29 |
| | NHE Attack (Ours) | | **94.46** | **94.58** | 40.05 | 34.14 | **56.48** | **33.52** | 31.45 | **54.95** | **2.00** |

Table 4: Evaluation of test-time data poisoning under the RTTDP protocol for ImageNet-C. We report the attack success rate (higher the better) for each TTA method and the average ranking (lower the better) for each attack objective. * indicates that the method is modified to align with the RTTDP protocol.

| Attack Freq. | Attack Objective | Source | TENT | SAR | CoTTA | ROID | Avg. Err. (↑) | Avg. Rank (↓) |
|---|---|---|---|---|---|---|---|---|
| Uniform | No Attack | | 63.49 | 61.26 | 63.02 | 53.42 | 60.30 | 5.50 |
| | DIA* (Wu et al., 2023) | | 67.18 | 62.91 | 64.09 | 56.97 | 62.79 | 4.00 |
| | TePA* (Cong et al., 2023) | 82.08 | 75.36 | 64.90 | 62.84 | 59.78 | 65.72 | 3.00 |
| | MaxCE (Madry et al., 2018) | | 62.64 | 61.66 | **68.83** | **59.89** | 63.26 | 3.25 |
| | BLE Attack (Ours) | | 68.04 | 64.31 | 66.40 | 57.10 | 63.96 | 3.00 |
| | NHE Attack (Ours) | | **78.03** | **72.58** | 63.84 | 57.72 | **68.04** | **2.25** |
| Non-Uniform | No Attack | | 61.81 | 59.52 | 62.51 | 50.36 | 58.55 | 6.00 |
| | DIA* (Wu et al., 2023) | | 66.61 | 62.88 | 63.40 | 55.18 | 62.02 | 4.25 |
| | TePA* (Cong et al., 2023) | 81.98 | 74.98 | 62.31 | 62.78 | 59.29 | 64.84 | 3.50 |
| | MaxCE (Madry et al., 2018) | | 62.09 | 65.58 | **72.39** | **59.76** | 64.96 | 2.50 |
| | BLE Attack (Ours) | | 67.61 | 65.95 | 65.75 | 56.23 | 63.89 | 2.75 |
| | NHE Attack (Ours) | | **77.49** | **73.67** | 64.02 | 57.09 | **68.07** | **2.00** |

than "Uniform" attack. This is probably due to consecutive attack being more effective in misleading model's update.

## 5.3 ABLATION STUDY ON ATTACK MODULES

In this section, we ablate our proposed modules including the surrogate model, the feature consistency regularization and two attack objectives to demonstrate their their indispensable contribution to the final results. We conduct the experiments on CIFAR10-C and CIFAR100-C datasets as shown in Tab. 5. First, comparing the use of the source model v.s. surrogate model for generating poisoned data, the surrogate model consistently delivers superior results, often approaching or even slightly surpassing those obtained with the online model, regardless of the attack objective. It demonstrates the effectiveness of our proposed surrogate model that is leveraged for generating on-the-fly poisoned data. Second, our proposed NHE attack objective is effective though using source model and without feature consistency regularization, that could be attributed to the high entropy samples easily mislead the model update and cause strong perturbation to the source knowledge. Third, under the surrogate model or online model, both BLE and NHE are significantly improved with the help of feature

Table 5: The ablation study of our proposed modules under the RTTDP protocol on CIFAR10/100-C datasets.

| Attack Model | Attack Objective | Feat. Cons. Reg. | CIFAR10-C | | | | CIFAR100-C | | | |
|---|---|---|---|---|---|---|---|---|---|---|
| | | | TENT | EATA | SAR | AR (↑) | TENT | EATA | SAR | AR (↑) |
| Source Model | BLE | - | 19.26 | 18.82 | 19.80 | 19.29 | 34.21 | 31.62 | 32.17 | 32.67 |
| Source Model | BLE | ✓ | 38.09 | 23.28 | 21.86 | 27.74 | 67.11 | 34.57 | 33.78 | 45.15 |
| Source Model | NHE | - | 63.00 | 23.76 | 19.52 | 35.43 | 85.48 | 39.37 | 38.76 | 54.54 |
| Source Model | NHE | ✓ | 37.27 | 20.79 | 21.07 | 26.38 | 81.81 | 35.96 | 47.06 | 54.94 |
| Surrogate Model (Ours) | BLE | - | 20.22 | 20.16 | 19.71 | 20.03 | 38.71 | 31.87 | 31.95 | 34.18 |
| Surrogate Model (Ours) | BLE | ✓ | 54.07 | 45.20 | 26.80 | 42.02 | 73.93 | 47.30 | 43.25 | 54.83 |
| Surrogate Model (Ours) | NHE | - | 72.77 | 21.93 | 19.51 | 38.07 | 81.49 | 32.16 | 31.68 | 48.44 |
| Surrogate Model (Ours) | NHE | ✓ | 73.86 | 29.73 | 24.56 | 42.72 | 92.08 | 37.86 | 56.09 | 62.01 |
| Online Model | BLE | - | 25.14 | 25.50 | 19.71 | 23.45 | 42.09 | 32.82 | 32.05 | 35.65 |
| Online Model | BLE | ✓ | 56.75 | 52.32 | 27.38 | 45.48 | 77.92 | 49.91 | 44.01 | 57.28 |
| Online Model | NHE | - | 72.01 | 21.91 | 19.52 | 37.81 | 80.62 | 31.96 | 31.69 | 48.09 |
| Online Model | NHE | ✓ | 73.62 | 28.73 | 25.03 | 42.46 | 91.15 | 39.01 | 54.67 | 61.61 |

consistency regularization, empirically demonstrating the reasonableness and effectiveness of our method.

### 5.4 EXPLORING EFFECTIVE DEFENSE PRACTICES

In this section, we explore effective defense practices. We conduct an ablation study for each of the above practices on top of a simple entropy minimization baseline method (Min. Ent.). As seen in Tab. 6, we make the observation that without any hypothesized defense practice, directly minimizing entropy is very sensitive to poisoned data, especially the high entropy attack (NHE). When entropy thresholding (Ent. Thresh.) is applied, we observe a significant improvement in robustness under high entropy attack, suggesting rejecting high entropy testing samples from TTA is an effective defense practice. Furthermore, both data augmentation (Data Aug.) and exponential moving average update (EMA) are very effective defense practices. The former could perturb the testing sample towards non-adversarial direction while the latter prevents the model from updating too quickly, thus less sensitive to poisoned data. Finally, one might expect stochastic parameter restoration (Stoch. Resto.) to be an effective defense method. Despite exhibiting improved adversarial robustness alone, parameter restoration does not further improve the robustness when combined with other effective defense methods.

Table 6: Ablation study of hypothesized defense practices on CIFAR10-C dataset.

| Min. Ent. | Ent. Thresh. | Data Aug. | EMA Update | Stoch. Resto. | BLE | NHE |
|---|---|---|---|---|---|---|
| - | - | - | - | - | 43.81 | |
| ✓ | - | - | - | - | 54.07 | 73.86 |
| ✓ | ✓ | - | - | - | 46.24 | 35.86 |
| ✓ | ✓ | ✓ | - | - | 24.01 | 20.05 |
| ✓ | ✓ | ✓ | ✓ | - | 20.22 | 19.76 |
| - | - | - | - | ✓ | 29.68 | 50.68 |
| ✓ | ✓ | ✓ | ✓ | ✓ | 20.41 | 20.30 |

Table 7: The generalisation of our modules on CIFAR10-C.

| Attack Objective | TENT | EATA |
|---|---|---|
| DIA | 26.04 | 18.94 |
| DIA + Ours | 27.02 | 19.41 |
| TePA | 33.78 | 23.37 |
| TePA + Ours | 38.79 | 21.37 |
| MaxCE | 18.55 | 18.17 |
| MaxCE + Ours | 23.61 | 19.44 |

### 5.5 GENERALISATION OF OUR MODULES

In this section, to demonstrate the generalization of our method, we combine the existing attack objectives with our proposed modules including the surrogate model and feature consistency regularization. The comparisons are shown in Tab. 7. First, we can observe that DIA and MaxCE could get improved additional with our proposed method. Second, TePA could obtain improvement under TENT method but slightly degraded under EATA method. It could be that TePA using maximizing entropy as an attack objective, and with the help of the surrogate model, the poisoned data would have very high prediction entropy because the attack reference model is more approximate to the online model, but fail to pass the entropy threshold and class diverse weighting using in EATA. Overall, our proposed in-distribution attack could generalize to most of the existing attacking objectives.

## 6 CONCLUSION

In this work, we reviewed the assumptions adopted by existing works for generating poisoned data at test-time and propose a few criteria that define a more realistic test-time data poisoning. Specifically, we approach from the angles of attack transparency, access to other users' benign data, attack budget, and attack order. To craft realistic poisoned data, we proposed a grey-box in-distribution attack with attack objective tailored for TTA methods. Through extensive evaluations under the realistic evaluation protocol, we reveal that the adversarial risk of TTA method might be over estimated and, importantly, certain practices in TTA methods are empirically proven to be effective and should be considered for designing adversarial robust TTA methods in the future.

ACKNOWLEDGEMENTS

This research work is supported by the National Natural Science Foundation of China (NSFC) (Grant Number: 62106078), the Agency for Science, Technology and Research (A*STAR) under its MTC Programmatic Funds (Grant Number: M23L7b0021) and the Guangdong R&D key project of China (Grant Number: 2019B010155001). This work was done during Yongyi Su's attachment with Institute for Infocomm Research (I2R), funded by China Scholarship Council (CSC).

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

# A APPENDIX

## A.1 ADDITIONAL DETAILS FOR METHODOLOGY

### A.1.1 ILLUSTRATION OF TEST-TIME DATA POISONING BATCH SPLIT

We visualize the batch split under realistic test-time data poisoning in Fig. 3. We present the batch split scheme for both "Uniform" and "Non-Uniform" attack frequencies. Under "Uniform" attack frequency, the poisoned minibatch is uniformly presented in the test data stream while "Non-Uniform" attack protocol simulates the situation the adversary attacks the TTA model in a short period of time with a huge amount of poisoned data.

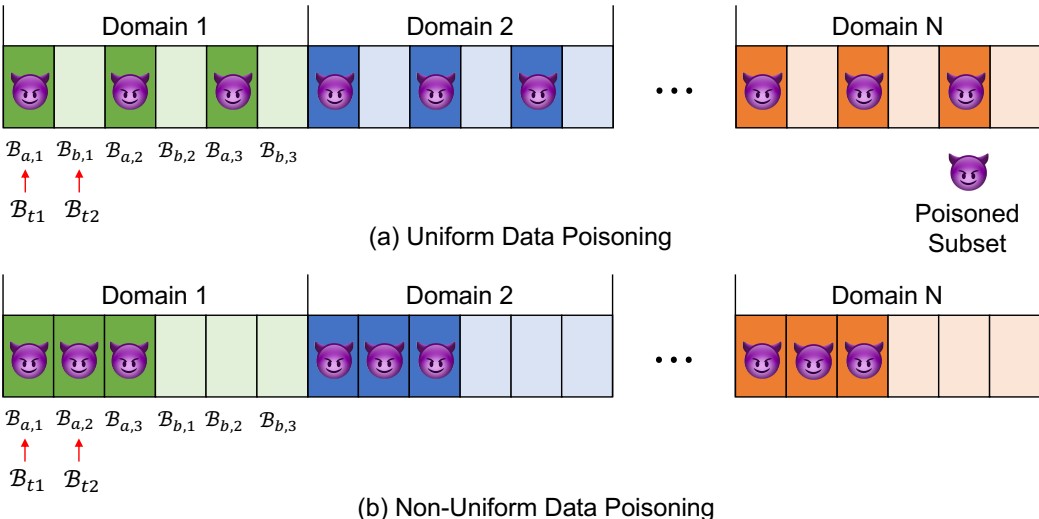

Figure 3: Illustration of test-time data poisoning batch split.

### A.1.2 GLOBAL OPTIOMAL LABEL MAPPING

For Balanced Low Entropy Attack, we need to obtain a label mapping which mainly addresses the following issues, i) maps the GT label to one wrong label; ii) the label mapping is bijective; iii) the sum of the mapping cost is the minimal. To achieve it, we first define a probability confusion between all class pairs as $C \in [0,1]^{K \times K}$. The probability confusion is updated in an exponentially moving average fashion using the current posterior predictions. The label mapping $M \in \{0,1\}^{K \times K}$ is then obtained by optimizing the following linear assignment problem. Efficient solver, e.g. Hungarian method, can be employed to solve this problem, as Eq. 9.

$$\hat{M} = arg \max_M \sum_k \sum_q C_{k,q} M_{k,q}$$

$$s.t. \quad M_{k,k} \neq 1, \quad \sum_q M_{k,q} = 1, \quad M \in \{0,1\}^{K \times K},$$

$$C_k^t = \beta C_k^t + (1 - \beta) \frac{\sum_{\tilde{x}_i \in \mathcal{B}_a} \mathbb{1}(y_i = k) \cdot h(\tilde{x}_i)}{\sum_{x_i \in \mathcal{B}_a} h(x_i)}$$

(9)

Here, we also provide the pseudo code of the implementation of BLE Attack, as follow,

```
def attack_objective(self, x, y):
    // x: (B, K) the predicted logit of poisoned data
    // y: (B, )  the ground true label of poisoned data
    // return the attack loss value
```

```
with torch.no_grad():

    // EMA update C^t_k
    curr_prob_term = scatter_mean(x.softmax(1), y[:, None],
    ↪  dim=0,
    ↪  out=torch.zeros_like(self.class_wise_momentum_prob))
    new_ema_prob = self.class_wise_momentum_prob.clone()
    new_ema_prob[y.unique()] = self.momentum_coefficient *
    ↪  new_ema_prob[y.unique()] + (1 -
    ↪  self.momentum_coefficient) *
    ↪  curr_prob_term[y.unique()]

    new_ema_prob_select = new_ema_prob.clone()
    diag_mask =
    ↪  torch.diag(torch.ones(new_ema_prob_select.shape[0]))
    new_ema_prob_select[diag_mask.bool()] = 0.

    // Find the global optimal mapping M
    label_mapping = y.new_zeros(new_ema_prob_select.shape[0],
    ↪  dtype=torch.long)
    for i in range(new_ema_prob_select.shape[0]):
        biased_prob, biased_class =
        ↪  F.normalize(new_ema_prob_select, dim=-1,
        ↪  p=1).max(dim=-1)
        max_item = biased_prob.argmax(dim=-1)
        label_mapping[max_item] = biased_class[max_item]
        new_ema_prob_select[max_item, :] = 0.
        new_ema_prob_select[:, biased_class[max_item]] = 0.

// BLE attack loss
loss = F.cross_entropy(x, label_mapping[y])
self.current_prob = new_ema_prob.detach()
return loss
```

### A.1.3  DETAILS OF DEFENSE PRACTICES

Here, we provide the details about the defense practices evaluated in Tab. 6 of the main text.

- **Entropy Thresholding** is implemented through filtering the entropy below $0.05 * log(K)$, where K is the class number of the dataset.
- **Data Augmentation** performs the data augmentation used into CoTTA over the input samples and constrain the consistent predictions between the augmented samples and the corresponding original samples.
- **EMA Update** module allow us to maintain an exponentially moving average updated model to generate robust predictions and used to supervise the online model update, and the EMA momentum is 0.999.
- **Stochastic Parameter Restoration** is implemented as the module used in CoTTA. We randomly reset the network weights to source model weights with probability $p$ and $p$ is set to 0.01.

### A.1.4  DISTINCTION BETWEEN RTTDP AND TePA

We would like to highlight the key differences between our proposed RTTDP and TePA protocols as follows,

TePA (Cong et al., 2023) employs a fixed surrogate model before test-time adaptation begins for generating poisoning, which qualifies the method as an offline method. The surrogate model is obtained by training a separate model (different architecture from the target model) using the same

source dataset. For example, on TTA for CIFAR10-C, if the target model, i.e. the model deployed for inference and is subject to test-time adaptation, is ResNet18, TePA employs VGG-11 as the surrogate model and trains VGG-11 on the same source training dataset (CIFAR10 clean training set). This is evidenced from the source code released by official repository [1] and the descriptions in TePA "we assume that the adversary has background knowledge of the distribution of the target model's training dataset. This knowledge allows the adversary to construct a surrogate model with a similar distribution dataset".

TePA employs the fixed surrogate model to generate poisoned dataset $x'$. Then generated poisoned dataset is fed to test-time adaptation to update model weights. Afterwards, TTA is further conducted on clean testing data for model update and performance evaluation. The segregation of data poisoning and TTA steps further support the claim that TePA should be classified as an offline approach.

Finally, to ensure fair comparison between TePA with our proposed methods under RTTDP protocol, TePA could be adapted to online fashion and we made such an adaptation to TePA for comparison in Tab. 2, Tab. 3 and Tab. 4 of the main text. Specifically, we use TePA to generate poisoning against the initial surrogate model and inject the generated poisoning into the testing data stream, i.e. placing poisoning in between benign testing batches. In this way, poisoning will affect TTA in an online fashion. We believe this is the most fair way to compare RTTDP with TePA.

### A.1.5   MORE ANALYSIS AND DERIVATIONS ABOUT THE OPTIMIZATION OBJECTIVE

Here, we discuss the transition from the original bi-level optimization objective (Eq. 2) to our proposed single-level optimization objective (Eq. 4) with a feature consistency constraint.

The original optimization objective for test-time data poisoning is formulated as a bi-level optimization problem, as shown below (equivalent to the meaning of Eq. 2):

$$\mathcal{B}_a = \arg \min_{\mathcal{B}_a} E_{(x,y) \in \mathcal{B}_b} \left[ \mathcal{L}_{atk}(h(x; \theta_t^*(\mathcal{B}_a \cup \mathcal{B}_b)), y) \right]$$
$$s.t. \ \theta_t^*(\mathcal{B}_a \cup \mathcal{B}_b) = \arg \min_{\theta_t} \mathcal{L}_{tta}(h(\mathcal{B}_a \cup \mathcal{B}_b; \theta)) \tag{10}$$

The above optimization involves an inner loop where the model adapts to test samples, including poisoned and benign samples, and an outer loop to optimize the attack objective. This structure is computationally intensive and impractical under the constraints of the RTTDP setting. To address this, we provide a detailed step-by-step derivation and explanation below.

**1. Discarding the Inner Optimization:** In DIA (Wu et al., 2023), the inner optimization is approximated by assuming $\theta_t^* \approx \theta_t$, where $\theta_t^*$ represents the parameters after a full adaptation step, and $\theta_t$ represents the current parameters. This approximation is justified as TTA models typically update minimally during a single minibatch iteration, resulting in minor perturbations to $\theta_t$. Thus, the approximation retains practical relevance while simplifying the problem. The formula is derived as (**this is also the DIA's objective**),

$$\mathcal{B}_a = \arg \min_{\mathcal{B}_a} E_{(x,y) \in \mathcal{B}_b} \left[ \mathcal{L}_{atk}(h(x; \theta_t(\mathcal{B}_a \cup \mathcal{B}_b)), y) \right] \tag{11}$$

**2. Surrogate Model for Online Parameters:** In the RTTDP protocol, direct access to online model parameters $\theta_t$ is unrealistic. Instead, we replace $\theta_t$ with the surrogate model parameters $\hat{\theta}_t$, which are accessible and trained to approximate the online model's behavior.

$$\mathcal{B}_a = \arg \min_{\mathcal{B}_a} E_{(x,y) \in \mathcal{B}_b} \left[ \mathcal{L}_{atk}(h(x; \hat{\theta}_t(\mathcal{B}_a \cup \mathcal{B}_b)), y) \right] \tag{12}$$

**3. Removing the access to $\mathcal{B}_b$:** In the RTTDP protocol, the adversary is prohibited from observing benign users' samples when generating poisoned samples. Consequently, the $\mathcal{B}_b$ term is excluded from the optimization objective. In the main text, we introduce to leverage $\mathcal{B}_{ab}$ to replace $\mathcal{B}_b$, where $\mathcal{B}_{ab}$ represents the adversary benign samples before they are poisoned.

---

[1] https://github.com/tianshuocong/TePA

$$\mathcal{B}_a = \arg\min_{\mathcal{B}_a} E_{(x,y)\in\mathcal{B}_{ab}} \left[ \mathcal{L}_{atk}(h(x; \hat{\theta}_t(\mathcal{B}_a \cup \mathcal{B}_{ab})), y) \right] \tag{13}$$

where $\hat{\theta}_t(\mathcal{B}_a \cup \mathcal{B}_b)$ indicates forwarding $\mathcal{B}_a \cup \mathcal{B}_b$ to update the BN statistics. This objective would lead to a trivial solution that $\mathcal{B}_a$ is effective only for the current $\mathcal{B}_{ab}$ data through easily introducing biased normalization in each BN layer, and it has little effect while $\mathcal{B}_a$ and $\mathcal{B}_{ab}$ are in seperated batch. Therefore, it would waste a half of attack query budget for forwarding these poisoned samples (the benign samples take up half of the batch size).

**4. Introducing a feature consistency constraint to improve query utilization:** In the main text, we observed the feature distributions of $\mathcal{B}_a$ and $\mathcal{B}_{ab}$ and found out that they obviously do not overlap, so we introduced feature consistency constraint to regularize their distributions according to the PAC learning framework in order to merge the two subsets into a single one, and to improve the utilization of the poisoned data query. The final objective is derived as follows, where $P_a$ and $P_{ab}$ are the shallow feature distributions of $\mathcal{B}_a$ and $\mathcal{B}_{ab}$, respectively.

$$\mathcal{B}_a = \arg\min_{\mathcal{B}_a} E_{(x,y)\in\mathcal{B}_a} \left[ \mathcal{L}_{atk}(h(x; \hat{\theta}_t(\mathcal{B}_a)), y) \right], \ \ s.t. \ \ D(P_a, P_{ab}) = 0 \tag{14}$$

To this end, we can fully utilize the budget of all poisoned data query to generate poisoned data. The experimental results show that the attack performance of the attack objective will be significantly improved after using this regularization term.

**In-distribution Attack Objective from a TTA Perspective:** Our proposed objective leverages the dependence of TTA models on self-training mechanisms, which aim to maximize confidence on pseudo-labels for adaptation. When the TTA model adapts to poisoned samples, it learns and reinforces incorrect associations. This creates a vulnerability, as future test samples with similar shallow feature distributions are more likely to be misclassified by the online model. Since TTA methods iteratively adapt using incoming test samples, our approach leverages this dependency to propagate the error induced by poisoned samples throughout the adaptation process.

### A.1.6 DETAILED POISONING & TRAINING ALGORITHM

We present the overall algorithm for generating poisoned data and surrogate model update in Alg. 1

### A.2 EXPERIMENTAL SETUPS OF DIA, TEPA AND RTTDP

We revisit the experimental setups of the previous methods, i.e. DIA (Wu et al., 2023) and TePA Cong et al. (2023), explain the differences under our RTTDP protocol, and justify the adaptations we made to ensure fair comparisons.

**Commonalities among the different protocols**: The three protocols, TePA, DIA, and RTTDP, share several overarching goals and assumptions. First, all protocols aim to evaluate the adversarial risks posed to Test-Time Adaptation (TTA) by injecting poisoned samples into the test data stream. Second, all protocols allow the adversary to obtain the source model, since the source model is usually the well-known pre-trained model, e.g., ImageNet pre-trained ResNet, and the open-source foundation model, e.g., DINOv2, SAM.

**Key Differences among Protocols**: Despite sharing some commonalities, the protocols diverge significantly in their attack setups.

In the TePA protocol, poisoned samples are generated by maximizing the entropy of the adversary's crafted samples with respect to the source model's predictions. These poisoned samples are injected into the TTA pipeline before any benign users' samples are processed, simulating an offline attack scenario. However, this approach is unrealistic in real-world settings, where adversaries cannot fully control the sequence of test samples in advance.

In contrast, the DIA protocol generates poisoned samples by optimizing them to maximize the cross-entropy loss of benign samples belonging to other users. DIA assumes direct access to the online model's parameters and the ability to observe other users' benign samples. Poisoned samples

**Algorithm 1:** The pipeline of RTTDP

**input** : A minibatch of testing samples $\mathcal{B}_t = \mathcal{B}_{ab} \cup \mathcal{B}_b$, where $\mathcal{B}_{ab}$ is the adversary benign subset that is preparing for crafting poisoned data and $\mathcal{B}_b$ is other users' benign subset.

Test-time adaptation model: $h(x; \theta_t)$,

Surrogate model used by adversary: $h(x; \hat{\theta}_t)$.

Attack Objective: $\mathcal{L}_{atk}$

// generate attack samples through surrogate model.

**if** $\mathcal{B}_a \neq \emptyset$ **then**

    initialize $\epsilon = \{0\}^{B \times H \times W \times 3}$, where $B = |\mathcal{B}_{ab}|$.

    initialize $\lambda = \{0\}^L$, where $L$ is the number of feature layers.

    **for** $i := 1$ **to** $40$ **do**

        $\mathcal{B}_a = \{\tilde{x}_i;\ \tilde{x}_i = x_i + \epsilon_i\}$

        calculate the attack loss: $loss_1 = \frac{1}{|\mathcal{B}_a|} \sum_{\tilde{x}_i \in \mathcal{B}_a} \mathcal{L}_{atk}(\tilde{x}_i)$.

        obtain all feature maps $\{\tilde{z}_i^l\}_{l=1\cdots L}$ before the normalization layers.

        calculate the feature consistency regularization term as Eq. 6:

        $loss_2^l = \sum_{\tilde{x}_i \in \mathcal{B}_a} KLD(\mathcal{N}(\mu_i^l, \Sigma_i^l) || \mathcal{N}(\tilde{\mu}_i^l, \tilde{\Sigma}_i^l))$

        construct the final optimized objective as Eq. 6:

        $\mathcal{L} = loss_1 + \frac{1}{L} \sum_l^L \lambda_l \cdot loss_2^l$

        update the adversarial noise:

        $\epsilon' = \epsilon - \alpha * sign\left[\nabla_\epsilon \mathcal{L}\right]$, where $\alpha$ is PGD attack step size of $0.01$.

        $\epsilon_i = clamp(x_i + \epsilon_i', 0, 1) - x_i, \quad x_i \in \mathcal{B}_a$.

        update the $\lambda_l$:

        $\lambda_l = \lambda_l + 0.001 \cdot \nabla_{\lambda_l} \mathcal{L}$

// feed into TTA model and obtain the prediction.

$y_i = h(x_i, \theta_t), \quad x_i \in \mathcal{B}_t$

// update the surrogate model if $\mathcal{B}_a \neq \emptyset$.

**if** $\mathcal{B}_a \neq \emptyset$ **then**

    $\hat{\theta}_{t,0} = \hat{\theta}_t$

    **for** $j := 1$ **to** $iters$ **do**

        $p_i^a = h(x_i^a, \theta_t), \quad x_i^a \in \mathcal{B}_a$.

        $\hat{p}_i^a = h(x_i^a, \hat{\theta}_{t,j-1}), \quad x_i^a \in \mathcal{B}_a$.

        calculate the distillation loss $\mathcal{L}_{dist}$ as Eq. 1.

        $\hat{\theta}_{t,j} = \hat{\theta}_{t,j-1} - lr * \nabla_{\hat{\theta}} \mathcal{L}_{dist}$

    $\hat{\theta}_{t+1} = \hat{\theta}_{t,iters}$.

are injected into the TTA pipeline alongside the corresponding benign users' samples. However, this protocol has significant limitations in realistic settings. In practice, adversaries typically lack access to or control over the online model's parameters. Additionally, it is highly improbable for adversaries to observe benign users' samples, let alone the validation samples required for optimizing poisoning objectives.

The proposed RTTDP protocol addresses these limitations by operating under more realistic assumptions. In RTTDP, the adversary neither has access to other users' benign samples nor the parameters of the online model. Instead, RTTDP employs a surrogate model, initialized as the source model, to generate poisoned samples. This surrogate model is iteratively updated based on feedback from previously injected poisoned samples. Poisoned samples are then injected into the TTA pipeline, either uniformly or non-uniformly, depending on the attack frequency in RTTDP protocol.

**Adaptations of Competing Methods to RTTDP protocol:** To ensure fair comparisons under RTTDP protocol, we made the following adjustments to the competing methods:

For TePA method, we preserved TePA's original poisoning objective, i.e. maximizing entropy, but adapted the poisoned data injection strategy from an offline manner to an online manner, i.e. placing poisoning in between benign testing batches according to RTTDP.

For DIA method, (1) Replacing Online Model Parameters: DIA's original objective relies on online model parameters, which are inaccessible in RTTDP. We replaced these parameters with the initial surrogate model, i.e. source model. (2) No Access to Benign Users' Samples for Optimization: DIA uses benign users' samples as optimization targets in its original setup. To meet RTTDP's constraints, we split $\mathcal{B}_{ab}$ into two equal subsets, $\mathcal{B}_{ab}^p$ and $\mathcal{B}_{ab}^b$, with a 1:1 ratio. We then generate poisoned samples $\mathcal{B}_a^p$ by maximizing the cross-entropy loss of $\mathcal{B}_{ab}^b$. The specific formula can be found in Eq. 15.

### A.3 ADDITIONAL DETAILS FOR EXPERIMENT

**Benchmark TTA Methods**: We investigate several state-of-the-art TTA methods under our RTTDP protocol to evaluate their adversarial robustness. **Source** serves as the baseline for inference performance without adaptation. **TENT** (Wang et al., 2020) updates BN parameters through minimizing entropy. **RPL** (Rusak et al., 2022) performs self-training with a generalized cross-entropy (GCE) loss, which aids in more robust adaptation under label noise. **EATA** (Niu et al., 2022) minimizes entropy with the Fisher regularization term to prevent forgetting knowledge from the source domain. **TTAC** (Su et al., 2022) adapts all backbone parameters by jointly optimizing global and class-wise distribution alignment with the source distribution. **SAR** (Niu et al., 2023) updates BN parameters with a sharpness-aware optimizer to filter out noisy labels and help escape local minima. **CoTTA** (Wang et al., 2022) leverages a teacher-student structure, optimizing all parameters of the student model and updating the teacher model using an exponential moving average. To better prevent forgetting during continual adaptation, it incorporates parameter random resetting and data augmentation methods. **ROID** (Marsden et al., 2024) updates BN parameters with loss of self-label refinement (SLR) weighed by certainty and diversity while continually weighting the online model and the source model to prevent forgetting.

**Evaluation Protocol**: We devise a evaluation plan respecting the realistic test-time data poisoning criteria. First, we investigate the frequency of injecting poisoned data. The "**Uniform**" scheme indicates that the poisoned minibatch is uniformly present in the test data stream, simulating the scenario that the adversary is periodically injecting the poisoned data. We further evaluate "**Non-Uniform**" scheme by allowing the adversary to concentrate the attack budget within a short period of time. We fix the overall attack budget as $r = \frac{|\mathcal{B}_a|}{|\mathcal{B}_a| + |\mathcal{B}_b|}$ throughout the experiments. We report the attack success rate as the evaluation metric, which is measured as the percentage of misclassified benign samples in the benign subset $\mathcal{B}_b$. Additionally, for easier comparison, we also calculate the average error rate (higher the better) and the average ranking (lower the better) for each poisoning method.

**Hyperparameters**: For all competing methods, we employ the 40 steps $L_\infty$ PGD attack to generate poisoned data. The maximum perturbation budget $b$ is 0.3 and the attack step size is 0.01. Additionally, within each PGD iteration, we update $\lambda_l$ via $\lambda_l' = \lambda_l + 0.001 \cdot \nabla_{\lambda_l} \mathcal{L}$ for Eq. 6, where $\lambda_l$ is initialized as zero before PGD attack. Unless otherwise noted, the overall attack budget $r$ is 50% throughout the

experiment. For surrogate model distillation module, we adopt SGD optimizer with 0.1 learning rate for 10 iterations to update the surrogate model in each update stage.

**Implementation Details**: For a fair comparison, we implement various data poisoning methods within a unified poisoning framework. This framework utilizes a 40-step Projected Gradient Descent (PGD Boyd & Vandenberghe (2004)) optimization process tailored to the respective objectives of each method. The poisoned samples generated are then injected into the TTA pipeline in an online manner, adhering to the RTTDP protocol. Specifically, the respective objectives of different competing poisoning methods are shown as follows,

- DIA (Wu et al., 2023):

$$\mathcal{B}_a^p = \arg \min_{\mathcal{B}_a^p} E_{(x,y)\in\mathcal{B}_{ab}^b} \left[ -CrossEntropyLoss(h(x; \hat{\theta}_0(\mathcal{B}_{ab}^b \cup \mathcal{B}_a^p)), y) \right] \quad (15)$$

- TePA (Cong et al., 2023):

$$\mathcal{B}_a = \arg \min_{\mathcal{B}_a} E_{(x,y)\in\mathcal{B}_a} \left[ -Entropy(h(x; \hat{\theta}_0(\mathcal{B}_a))) \right] \quad (16)$$

- MaxCE (Madry et al., 2018):

$$\mathcal{B}_a = \arg \min_{\mathcal{B}_a} E_{(x,y)\in\mathcal{B}_a} \left[ -CrossEntropyLoss(h(x; \hat{\theta}_0(\mathcal{B}_a)), y) \right] \quad (17)$$

- Unlearnable Examples (Huang et al., 2021):

$$\mathcal{B}_a = \arg \min_{\mathcal{B}_a} E_{(x,y)\in\mathcal{B}_a} \left[ CrossEntropyLoss(h(x; \theta_{init}(\mathcal{B}_a)), y) \right] \quad (18)$$

- Adversarial Poisoning (Fowl et al., 2021):

$$\mathcal{B}_a = \arg \min_{\mathcal{B}_a} E_{(x,y)\in\mathcal{B}_a} \left[ CrossEntropyLoss(h; \hat{\theta}_0(\mathcal{B}_a), \hat{y}) \right], \text{where } \hat{y} = (y+1)\%K. \quad (19)$$

- NHE Attack (Ours):

$$\mathcal{B}_a = \arg \min_{\mathcal{B}_a} \max_{\lambda} E_{(x,y)\in\mathcal{B}_a} \left[ \mathcal{L}_{atk}^{NHE}(x; \hat{\theta}_t(\mathcal{B}_a)) + \lambda \mathcal{L}_{reg} \right] \quad (20)$$

- BLE Attack (Ours):

$$\mathcal{B}_a = \arg \min_{\mathcal{B}_a} \max_{\lambda} E_{(x,y)\in\mathcal{B}_a} \left[ \mathcal{L}_{atk}^{BLE}(x; \hat{\theta}_t(\mathcal{B}_a)) + \lambda \mathcal{L}_{reg} \right] \quad (21)$$

where $\hat{\theta}_0$ indicates the initial surrogate model parameters, $\theta_{init}$ indicates the randomly initialized parameters and $K$ is the number of the category in the dataset. Since under the RTTDP protocol the real-time model parameters are unavailable to access, we leverage the initial surrogate model, to replace the online model they might have used in their original paper, as the threat model for the competing methods. The surrogate model is initialized as source model. For our proposed methods, we employ the proposed surrogate model distillation module to update the surrogate model, and during each PGD iteration, the variables $\mathcal{B}_a$ and $\lambda$ are updated simultaneously using gradient descent for $\mathcal{B}_a$ and gradient ascent for $\lambda$, respectively. More details about our proposed methods can be found in Alg. 1.

## A.4 Additional Empirical Analysis

### A.4.1 Comparison with Adversarial Attack Methods

Adversarial attack methods are designed to generate perturbations on input samples to mislead the model into making incorrect predictions. Here, we aim to investigate whether the adversarial effects of poisoned samples, generated using advanced adversarial attack methods (Madry et al., 2018; Croce & Hein, 2020; Chen et al., 2022), can be effectively transferred to benign users' samples under the RTTDP protocol.

We conduct the experiments on CIFAR10-C and ImageNet-C datasets with a Uniform attack frequency under our proposed RTTDP protocol. The results are shown in Tab. 8 and Tab. 9. We make the following observations. (i) The objectives of adversarial attack methods and data poisoning methods differ fundamentally. Adversarial attack methods focus on generating adversarial noise to mislead model predictions on the perturbed test samples. In contrast, data poisoning methods aim to inject carefully crafted poisoned samples to degrade the model's performance on subsequent benign samples after adaptation. (ii) While AutoAttack (Croce & Hein, 2020) represents a more advanced adversarial attack method, its performance is inferior to that of MaxCE-PGD (Madry et al., 2018) on ImageNet-C. (iii) Furthermore, certain complex adversarial attack methods, such as GMSA-MIN and GMSA-AVG (Chen et al., 2022), require generating adversarial perturbations separately for each class, a process that incurs substantial computational costs and limits scalability (which is why these methods are excluded from comparison on ImageNet-C), and still fall short compared to the efficacy of our proposed data poisoning methods.

Table 8: Comparison with different adversarial attack methods with our proposed data poisoning methods on CIFAR10-C dataset under the RTTDP protocol.

| Attack Objective | TENT | EATA | SAR | ROID | Avg |
|---|---|---|---|---|---|
| NoAttack | 19.72 | 18.03 | 18.94 | 16.37 | 18.27 |
| MaxCE-PGD (Madry et al., 2018) | 18.55 | 18.17 | 19.50 | 18.57 | 18.70 |
| AutoAttack (Croce & Hein, 2020) | 26.29 | 19.12 | 19.56 | 18.67 | 20.91 |
| GMSA-MIN (Chen et al., 2022) | 35.92 | 22.78 | 19.99 | 18.65 | 24.33 |
| GMSA-AVG (Chen et al., 2022) | 38.80 | 21.89 | 19.95 | 18.51 | 24.79 |
| BLE Attack (Ours) | 54.07 | **45.20** | **26.80** | **19.06** | 36.28 |
| NHE Attack (Ours) | **73.86** | 29.73 | 24.56 | 17.00 | **36.29** |

Table 9: Comparison with different adversarial attack methods with our proposed data poisoning methods on ImageNet-C dataset under the RTTDP protocol.

| Attack Objective | TENT | SAR | CoTTA | ROID | Avg |
|---|---|---|---|---|---|
| NoAttack | 63.49 | 61.26 | 63.02 | 53.42 | 60.30 |
| MaxCE-PGD (Madry et al., 2018) | 62.64 | 61.66 | **68.83** | **59.89** | 63.26 |
| AutoAttack (Croce & Hein, 2020) | 64.48 | 61.42 | 63.03 | 54.78 | 60.93 |
| BLE Attack (Ours) | 68.04 | 64.31 | 66.40 | 57.10 | 63.96 |
| NHE Attack (Ours) | **78.03** | **72.58** | 66.40 | 57.72 | **68.68** |

### A.4.2 ABLATION STUDY ON QUERY COUNTS

Regarding varying query attempts, we add an additional evaluation as follows. Nonetheless, we want to highlight that the query attempts do not have to be limited for our method because all queries are submitted to the surrogate model rather than the online model. More queries simply makes generating poisoning slower. In this study, we vary the query steps from 10 to 60 for the projected gradient descent optimization (Boyd & Vandenberghe, 2004). We evaluate varying attack query counts for two TTA methods under their respective strongest attack objectives. The results in the Tab. 10 are obtained on CIFAR10-C dataset with a Uniform attack frequency. We make the following observations. (i) Increasing the number of queries could improve the performance at a low query budget. (ii) When the budget is increased to beyond 40 queries, the performance saturates. We draw the conclusion that allowing sufficient queries to the surrogate model is necessary for generating effective data poisoning, and, importantly, this procedure will not create alert to the online model.

### A.4.3 ANALYSIS ON SYMMETRIC KLD USED FOR DISTILLING SURROGATE MODEL

In this work, we adopt the common practice of symmetrizing the Kullback-Leibler Divergence (KLD) to ensure balanced alignment between distributions in the surrogate model distillation. Following the

Table 10: The ablation study on query counts. These results are obtained on CIFAR10-C dataset with a Uniform attack frequency under RTTDP protocol. We choose 40 queries throughout the experiments.

| TTA Method | 10 | 20 | 30 | **40** | 50 | 60 |
|---|---|---|---|---|---|---|
| TENT (NHE Attack) | 66.95 | 74.36 | 74.34 | 73.86 | 73.51 | 73.66 |
| EATA (BLE Attack) | 35.73 | 39.70 | 42.36 | 45.20 | 45.99 | 45.89 |

definitions provided in the main text, the forward KLD is expressed as $KLD(h(x_i; \theta_t) \| h(x_i; \hat{\theta}_t))$, while the reverse KLD is defined as $KLD(h(x_i; \hat{\theta}_t) \| h(x_i; \theta_t))$.

Forward KLD emphasizes penalizing discrepancies where the distilled (surrogate) model $\hat{\theta}_t$ assigns low probability to samples that the source (real-time target) model $\theta$ deems important. It encourages the distilled model to mimic the behavior of the target model by focusing on areas of high confidence in $\theta$'s posterior.

Reverse KLD, in contrast, focuses on matching $\theta$'s predictions where $\hat{\theta}$ assigns high probabilities. This can result in sharper, more focused distributions but might dismiss less probable regions of $\theta$'s posterior.

The symmetric KLD balances the above two objectives. The forward KLD may be more suitable when surrogate model is significantly smaller than the target model and the objective is to allow the surrogate model to mimic the target model's certainty. When the surrogate model is of the same capacity with target model, using the symmetric KLD may better align the two models in both high confident and low confident predictions. In this work, the capacity of surrogate is similar to target model. Thus, we hypothesize that the symmetric KLD could be better.

We further use empirical observations in the Tab. 11 below to support the hypothesis. With symmetric KLD the performance is slightly better than using the forward KLD.

Table 11: Comparison between Symmetric KLD and Forward KLD used for surrogate model distillation. These results are obtained on CIFAR10-C dataset with a Uniform attack frequency under RTTDP protocol.

| TTA Method | Symmetric KLD | $KLD(h(x_i; \theta_t) \| h(x_i; \hat{\theta}_t))$ |
|---|---|---|
| TENT (NHE Attack) | 73.86 | **74.35** |
| EATA (BLE Attack) | **45.20** | 43.99 |
| SAR (BLE Attack) | **26.80** | 26.35 |

Nevertheless, we do acknowledge that both symmetric KLD and forward KLD give competitive results. The choice depends on computation affordability and empirical observations.

### A.4.4 DIFFERENT ATTACK BUDGETS

We further evaluate the effectiveness of proposed poisoning approach under different attack budgets. Specifically, we evaluated at $r = 0.1$, $r = 0.2$ and $r = 0.5$. We clearly observe that both high entropy and low entropy attacks are effective regardless of attack budgets.

### A.4.5 VISUALIZATION OF POISONED SAMPLES

We visualize selected samples before and after test-time data poisoning in Fig. 4. The high corruption level makes the adversarial noise less noticeable, suggesting the poisoned data could even evade human inspection.

Table 12: Comparing the attack performance of test-time data poisoning under different attack budgets.

| TTA | Attack Obj. | 0.1 | 0.2 | 0.5 |
|---|---|---|---|---|
| TENT | No Attack | 20.72 | 20.39 | 19.72 |
| | BLE Attack | 22.44 | 27.60 | 54.07 |
| | NHE Attack | 39.20 | 62.19 | 73.86 |
| EATA | No Attack | 17.99 | 17.76 | 18.03 |
| | BLE Attack | 22.20 | 28.29 | 45.20 |
| | NHE Attack | 19.59 | 20.10 | 29.73 |
| SAR | No Attack | 18.95 | 18.90 | 18.94 |
| | BLE Attack | 19.90 | 21.30 | 26.80 |
| | NHE Attack | 19.33 | 20.74 | 24.56 |

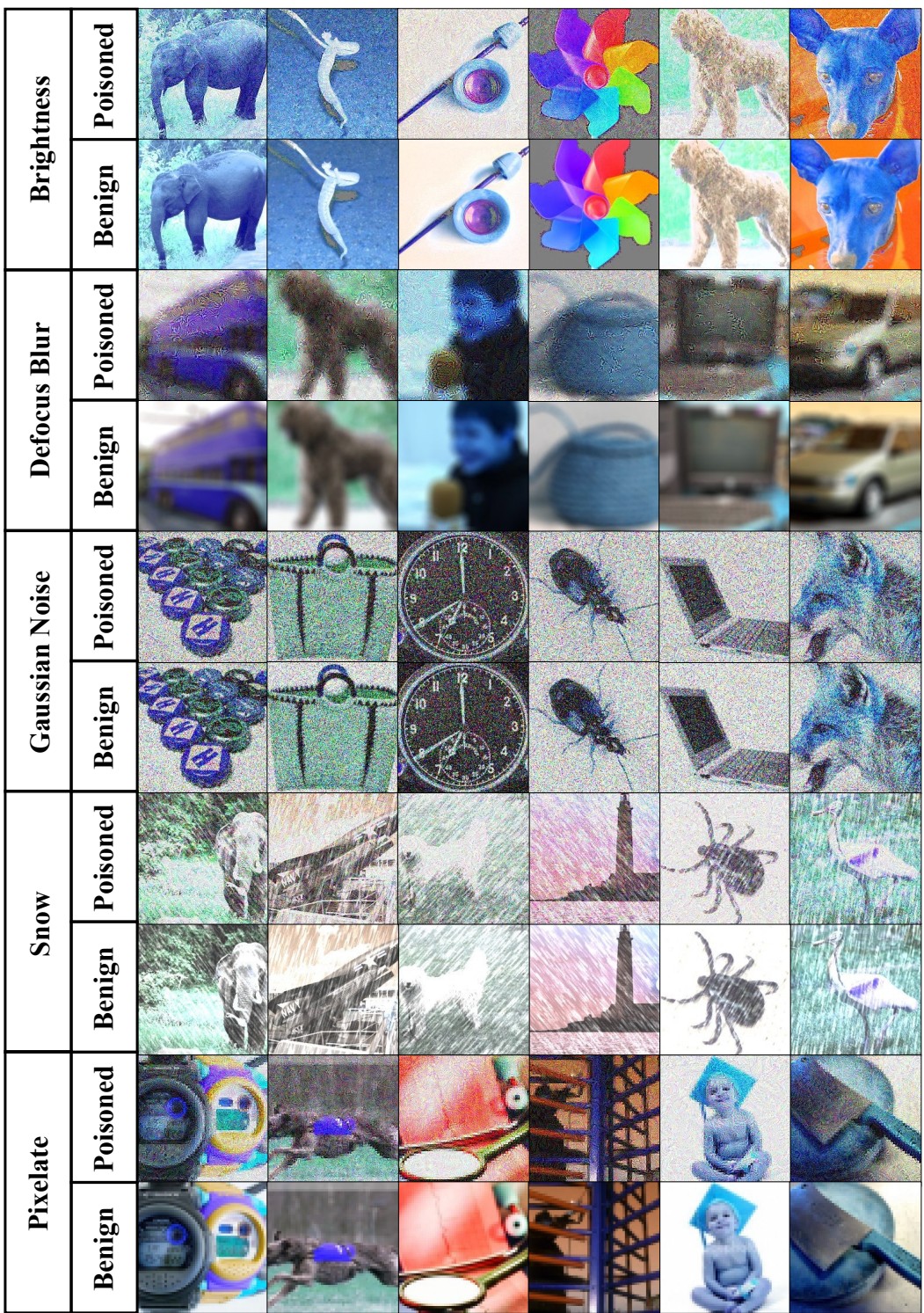

Figure 4: Visualizing of selected samples before and after test-time data poisoning.

