# OpenReview forum: "On the Adversarial Risk of Test Time Adaptation: An Investigation into Realistic Test-Time Data Poisoning"
_ICLR.cc/2025/Conference — ICLR 2025 Poster_

### Official Review · Reviewer_hmy9 · 2024-10-31

**Soundness:** 3
**Presentation:** 2
**Contribution:** 2
**Rating:** 5
**Confidence:** 4

**Summary:**

This paper explores test-time data poisoning (TTDP) under realistic conditions, proposing a grey-box attack that reflects practical scenarios where adversaries lack full access to benign data and real-time model updates. The study introduces two attack objectives based on entropy manipulation to effectively poison data while remaining within realistic constraints. Through extensive experimentation on state-of-the-art test-time adaptation (TTA) methods, the authors assess the vulnerability of these methods and propose feature consistency regularization as a countermeasure. This work highlights the persistent adversarial risks in TTA setups under real-world scenarios.

**Strengths:**

- The study is well-written;
- It covers an interesting setting that deserve much more attention from the security aspect.

**Weaknesses:**

While the study is well-written and formally structured, there are several areas where clarity and methodological rigor could be improved.

- Distinction between RTTDP and TePA in Table 1. In Table 1, the difference between RTTDP and TePA, particularly in terms of online adaptability, is not clear. It would be beneficial to clarify whether TePA is prevented from online applicability, thus highlighting RTTDP’s novelty.

- Inconsistent comparison between poisoning and adversarial attacks. The paper compares the proposed poisoning attack with MaxCE, an adversarial example attack applied at test time rather than during training. This inclusion creates confusion about the setting being considered, especially as MaxCE’s performance is close to that of the proposed attacks, despite being a simpler, suboptimal approach. Additionally, this inconsistency leads to general confusion throughout the paper regarding the exact threat model, setting, and practical cases under consideration. The title’s emphasis on “Realistic” test-time data poisoning also adds to this confusion, as data poisoning traditionally pertains to training rather than test inference. To clarify, the authors should explain that, while poisoning data are collected during test inference, they are subsequently used to update the model through adaptation techniques, which occurs as an offline process. Better descriptions and visualizations of the threat model, attack setting, and relevant practical scenarios would improve clarity and accurately convey the scope and applicability of the proposed approach.

- Not impactful results. MaxCE’s performance, although designed as an adversarial example attack rather than a poisoning attack, remains close to that of the proposed attacks, which raises questions about the actual impact and effectiveness of the proposed methods. The results do not demonstrate a clear or significant advantage of the new attack over MaxCE, suggesting that the proposed methods may lack the robustness or advancement intended. Furthermore, more sophisticated adversarial example attacks (e.g., PGD, C&W, and AutoAttack). I recommend including comparisons with these advanced adversarial attacks to provide a clearer perspective on whether the proposed approach offers meaningful improvements. Currently, the lack of a notable performance gap reduces the impact of the results, leaving it uncertain whether the proposed attacks represent a substantial advancement.

- Unsupported and unclear methodology. The transition from a bilevel optimization formulation to a single-level optimization problem lacks theoretical clarity. Such transitions are complex, and previous research has shown limitations in achieving optimal results through this method. Expanding on this approach and its theoretical basis would improve the rigor of the paper’s claims.

- Confusing experimental setup. The design choices in the "Benchmarking Poisoning Methods" section lack clarity. The authors do not specify how these choices impact prior works or how they differ from established benchmarking methods, such as DIA and TePA. This lack of clarity makes it difficult to understand the setup and its implications for reproducibility.

- Missing details on attack queries. The number of queries used by each attack is a critical factor, yet the experimental comparisons lack this information. Since query efficiency is a key point raised by the authors, detailing query usage across attacks would be essential for a fair comparison.


**Minor Points:**

- In Equation 1, the authors include two Kullback-Leibler Divergence (KLD) terms. To clarify their necessity, the authors should explain why both terms are required rather than just one, even if asymmetry in KLD is a factor. Providing justification in the text would enhance methodological clarity.

- I suggest the authors break Equation 2 into multiple lines to improve readability.

- Expanding the background discussion presented in Section 2 and related work with further background knowledge on adversarial examples, and on white-box, grey-box, and black-box threat models in the context of poisoning. The authors could reference relevant surveys that would help to clarify their threat model and position at the SoA and facilitate non-expert readers understand these distinctions.

- Figure 1 is overly technical and lacks clarity regarding the threat model and attacker strategy. Simplifying it could enhance its value as a conceptual overview.


**References**
- [AutoAttack] Reliable evaluation of adversarial robustness with an ensemble of diverse parameter-free attacks.
- [PGD] Towards deep learning models resistant to adversarial attacks
- [C&W] Towards evaluating the robustness of neural networks

**Questions:**

How does your approach transition from a bilevel optimization problem to a single-level one, and could you provide additional theoretical support for this methodology?

Can you elaborate on how your benchmarking setup differs from prior works and how these design choices may influence your results?

Given that MaxCE is primarily an adversarial example attack, what was the rationale behind including it in your comparisons, and have you considered additional comparisons with more advanced adversarial example attacks like PGD or AutoAttack?

---

> ### Author Response · Authors · 2024-11-24
> **Response to Reviewer hmy9 (Part I)**
>
> We highly appreciate the reviewer's efforts in providing valuable comments and constructive suggestions for improvement on our submission.
>
> **W1: Distinction between RTTDP and TePA in Table 1.**
>
> We would like to highlight the key differences between our proposed RTTDP and TePA protocols:
>
> - **Attack Order**: TePA follows an **offline attack order**, where poisoned data are injected into the TTA model before any benign test data. In contrast, RTTDP follows an **online attack order**, where poisoned data can be injected dynamically between batches of benign test data.
>
> - **Adversary Knowledge**: TePA assumes that the adversary has access to the model parameters when generating poisoned data, while RTTDP assumes that the adversary cannot access the real-time model parameters.
>
> - **Timing of Poisoned Data Injection**: RTTDP allows poisoned data to be injected at any time in the test data stream, making it compatible with real-time online scenarios. TePA, on the other hand, requires poisoned data to be introduced prior to any benign user data.
>
> - **Test Domain Setting**: RTTDP operates under the **continual test-time adaptation (CoTTA)** [1] setting, where the test data distribution changes gradually over time. However, TePA evaluates the poisoning effect under an **individual test domain adaptation**, which assumes a **static and single test data distribution**.
>
> [1] Continual Test-Time Domain Adaptation
>
> **W2(A): Inconsistent comparison between poisoning and adversarial attacks.**
>
> Thank you for your valuable feedback. We would like to clarify that in Tables 2 to 4 of the manuscript, all competing attack objectives, including DIA, TePA, and MaxCE, are implemented within the RTTDP protocol. All these methods generate poisoned data using the PGD attack. However, due to the lack of access to real-time model parameters under the RTTDP protocol, we use the source model as the threat model for comparison.
>
> Here are the specifics of the implemented methods:
>
> - **DIA:** We adapt the original DIA objective to our RTTDP framework by splitting $\mathcal{B} _{ab}$ into two equal subsets, $\mathcal{B} _{ab}^p$ and $\mathcal{B} _{ab}^b$, with a 1:1 ratio. We then generate poisoned samples $\mathcal{B} _a^p$ by maximizing the cross-entropy loss of $\mathcal{B} _{ab}^b$, as follows:
>
>     $$\mathcal{B} _a^p = \arg\min _{\mathcal{B} _a^p} E _{(x,y)\in\mathcal{B} _{ab}^b} \left[-\text{CrossEntropyLoss}\left(f(x, \theta _{\text{source}}(\mathcal{B} _{ab}^b \cup \mathcal{B} _a^p)), y\right)\right]$$
>
> - **TePA:** Following the original paper, we generate poisoned samples by maximizing the entropy of the samples, expressed as:
>
>     $$\mathcal{B} _a = \arg\min _{\mathcal{B} _a} E _{(x,y)\in\mathcal{B} _a} \left[-\text{Entropy}\left(f(x, \theta _{\text{source}}(\mathcal{B} _a))\right)\right]$$
>
> - **MaxCE:** We generate poisoned samples by maximizing the cross-entropy loss on the poisoned samples, as follows.PGD attack is employed to attack this loss and we are sorry for referring to the wrong reference (FGSM).
>
>     $$\mathcal{B} _a = \arg\min _{\mathcal{B} _a} E _{(x,y)\in\mathcal{B} _a} \left[-\text{CrossEntropyLoss}\left(f(x, \theta _{\text{source}}(\mathcal{B} _a)), y\right)\right]$$
>
> In terms of the optimization strategy, we employ the **PGD attack**, as detailed at line 342 in the manuscript. After generating the poisoned samples, they are injected into the TTA pipeline alongside the normal users' samples $\mathcal{B} _b$. The attack success rate is then evaluated on the benign users' samples.
>
> We will ensure this explanation is clarified in the **Benchmark Poisoning Methods** subsection of the revised manuscript. This should resolve any confusion between the different attack strategies and their implementation under RTTDP.
>
> Beyond using PGD attack, we also evaluated AutoAttack to generate strong adversarial samples with details presented in responts to W3\&Q3. Despite being strong as adversarial samples, samples generated by AutoAttack do not pose poisoning effect to TTA model.

---

> ### Author Response · Authors · 2024-11-24
> **Response to Reviewer hmy9 (Part II)**
>
> **W2(B): The further descriptions of Realistic Test-Time Data Poisoning.**
>
> Traditional data poisoning is typically an **offline process**, where the poisoned dataset is used to train a randomly initialized model over multiple epochs until convergence. In contrast, our proposed **RTTDP** introduces a novel **online test-time data poisoning** setting.
>
> RTTDP operates within a **Test-Time Adaptation (TTA)** [2] framework, which is an **online fine-tuning** method. TTA methods adapt a source pre-trained model to the test data distribution. Specifically, when a testing data batch is fed into the TTA model, it updates the online model using an **unsupervised loss** for a few iterations (usually one) with the current testing data batch. The model then instantly outputs predictions for this batch. In RTTDP, poisoned data are directly injected into the TTA model during the test phase, rather than being handled separately offline. **These poisoned samples are treated similarly to normal testing batches** and are used to update the online model for a single iteration. The objective is to investigate how these poisoned data affect the predictions of other benign samples.
>
> To improve clarity, we will elaborate on these points in the revised manuscript. This includes adding more details about the **preliminary concepts**, such as an introduction to **Test-Time Adaptation** and a comparison with **Traditional Data Poisoning** approaches. More details about the comparison with traditional data poisoning can be found in the response to Reviewer AP94 #W1.
>
> [2] Tent: Fully Test-time Adaptation by Entropy Minimization
>
>
> **W3&Q3: Not impactful results.**
>
> Thanks for your suggestions. We first would like to clarify that the MaxCE used into our experiments is also optimized via PGD attack and we apologize for making reference to the earlier paper in the original manuscript, where FGSM was proposed. The improvement based on MaxCE attack method is not marginal, especially for TENT, EATA and SAR. Furthermore, we supplement the experiment using a more advance adversarial attack method, i.e. AutoAttack, to generate the poisoned samples. The results on CIFAR10-C and ImageNet-C are shown below, and we make the following observations.
>
> - The goals of adversarial attack methods and data poisoning methods are significantly different. The adversarial attack methods aim to make their own samples predict incorrectly by corrupting the test samples with adversarial noise. However, data poisoning methods aim to inject some poisoned samples so that the online model updated on them would perform poorly on other benign samples.
> - AutoAttack, as a more advanced adversarial attack method, would produce worse results than MaxCE-PGD. We used code from the official AutoAttack repository and repeated the experiments to ensure the reliability and reproducibility of these results.
>
> We will supplement all the results of AutoAttack on the CIFAR10/100-C and ImageNet-C, as well as more implementation details about MaxCE in the camera-ready version.
>
>
> The results on CIFAR10-C dataset with `Uniform` attack frequency.
> |Attack Objective|TENT|EATA|SAR|ROID|Avg|
> |-|-|-|-|-|-|
> |MaxCE-PGD|18.55|18.17|19.50|18.57|18.70|
> |AutoAttack|26.29|19.12|19.56|18.67|20.91|
> |BLE Attack(Ours)|54.07|**45.20**|**26.80**|**19.06**|36.28|
> |NHE Attack(Ours)|**73.86**|29.73|24.56|17.00|36.29|
> |Our Best|**73.86**|**45.20**|**26.80**|**19.06**|**41.23**|
>
> The results on ImageNet-C dataset with `Uniform` attack frequency.
> |Attack Objective|TENT|SAR|CoTTA|ROID|Avg|
> |-|-|-|-|-|-|
> |MaxCE-PGD|62.64|61.66|**68.83**|**59.89**|63.26|
> |AutoAttack|64.48|61.42|63.03|54.78|60.93|
> |BLE Attack(Ours)|68.04|64.31|66.40|57.10|63.96|
> |NHE Attack(Ours)|**78.03**|**72.58**|63.84|57.72|68.04|
> |Our Best|**78.03**|**72.58**|66.40|57.72|**68.68**|
>
>
> **Why do we include MaxCE in our comparisons?**
>
> This is because we want to see if the adversarial effect can be transferred from the poisoned samples to the benign users' samples.
> In our experiment, we also compare with some common adversarial attack methods, such as MaxCE-PGD. For the adversarial attack method, we use it to generate the poisoned samples instead of directly modifying the samples of the benign users to achieve realism in the RTTDP protocol.
> Through comparisons, we observe that (i) the goal of adversarial attack is very different from that of our data poisoning; (ii) Figure 2(b) found that the samples generated by directly maximizing cross-entropy would produce a significant bias with the normal samples. This observation largely motivates us to introduce a feature constraint to transfer the adversarial effect from the poisoned samples to the other benign samples.
> Although some of the results in ImageNet-C may behave differently than we would have expected, we analyse that because the source results in ImageNet-C are so bad, the feature points are so dispersed in the feature space that it is difficult to approximate them by a Gaussian distribution.

---

> ### Author Response · Authors · 2024-11-24
> **Response to Reviewer hmy9 (Part III)**
>
> **W5&Q2: Confusing experimental setup.**
>
> Thank you for raising this concern. We would provide a clearer explanation of the design choices in our experimental setup for "Benchmarking Poisoning Methods." Below, we revisit the experimental setups of the original methods, explain the differences under our RTTDP protocol, and justify the adaptations we made to ensure fair comparisons. Additionally, to enhance reproducibility, we promise that we will release all source code, including our method and the competing methods, as soon as this manuscript will be accepted.
>
> **Commonalities among the different protocols:**
>
> The three protocols, **TePA**, **DIA**, and **RTTDP**, share several overarching goals and assumptions:
>
> - All protocols aim to evaluate the adversarial risks posed to **Test-Time Adaptation (TTA)** by injecting poisoned samples into the test data stream.
> - All protocols allow the adversary to obtain the source model, since the source model is usually the well-known pre-trained model, e.g., ImageNet pre-trained ResNet, and the open-source foundation model, e.g., DINOv2, SAM.
>
> **Key Differences Between Protocols:**
>
> Despite these commonalities, the protocols diverge significantly in their attack setups, as summarized below:
>
> 1. **TePA Protocol:**
>     - **Poisoning Objective:** TePA generates poisoned samples by maximizing the entropy of its own crafted samples with respect to the source model’s predictions.
>     - **Injection Strategy:** All poisoned samples are injected into the TTA pipeline **before any benign users’ samples are processed**. This approach simulates an offline attack, which is unrealistic in real-world scenarios where the adversary cannot fully control the sequence of test samples in advance.
>
> 2. **DIA Protocol:**
>    - **Poisoning Objective:** DIA optimizes poisoned samples to maximize the cross-entropy loss of other users’ benign samples. And DIA relies on direct access to **online model parameters** and the ability to observe **other benign users’ samples**.
>    - **Injection Strategy:** Poisoned samples are injected into the TTA pipeline alongside corresponding benign users’ samples.
>    - **Limitations for Realistic Settings:**
>      - **Online Model Access:** In practice, the adversary cannot access or modify the parameters of the online model.
>      - **Other Benign Users’ Data:** The adversary is unlikely to observe benign users’ samples, let alone validation samples required for optimizing poisoning objectives.
>
> 3. **RTTDP Protocol (Ours):**
>    - **Poisoning Objective:** RTTDP uses a surrogate model (initialized as the source model) to generate poisoned samples. This surrogate model is updated iteratively based on the feedback from previously injected poisoned samples.
>    - **Injection Strategy:** Poisoned samples are injected into the TTA pipeline either uniformly or non-uniformly:
>      - **Uniform:** Poisoned minibatches are evenly distributed across the test data stream.
>      - **Non-Uniform:** Poisoned minibatches are concentrated at specific points in the test data stream.
>    - **Realistic Assumptions:** RTTDP assumes the adversary has no access to the online model parameters or other users’ samples. This ensures a more realistic and practical attack scenario.
>
> **Adaptations of Competing Methods for RTTDP:**
>
> To ensure fair comparisons under RTTDP protocol, we made the following adjustments to the competing methods:
>
> 1. **TePA:**
>    We preserved TePA’s original poisoning objective but adapted the injection strategy to follow the **Uniform** or **Non-Uniform** attack frequency in RTTDP protocols.
>
> 2. **DIA:**
>    - **Replacing Online Model Parameters:** DIA’s original objective relies on online model parameters, which are inaccessible in RTTDP. We replaced these parameters with the source model parameters.
>    - **No Access to Benign Users' Samples for Optimization:** DIA uses benign users’ samples as optimization targets in its original setup. To meet RTTDP’s constraints, we split $\mathcal{B} _{ab}$ into two equal subsets, $\mathcal{B} _{ab}^p$ and $\mathcal{B} _{ab}^b$, with a 1:1 ratio. We then generate poisoned samples $\mathcal{B} _a^p$ by maximizing the cross-entropy loss of $\mathcal{B} _{ab}^b$. Details of this modification are provided in our response to **W2(A)**.
>
> **Justification of Design Choices:**
>
> - **Realism:** RTTDP reflects more realistic assumptions than TePA and DIA by removing unrealistic assumptions, such as online model access and visibility of other users’ samples.
> - **Fairness:** All competing methods are implemented under the RTTDP protocol, with necessary adaptations to maintain methodological integrity while adhering to RTTDP constraints.
> - **Reproducibility:** We will release the complete implementation of our proposed method and the adapted competing methods to ensure full transparency and reproducibility.

---

> ### Author Response · Authors · 2024-11-24
> **Response to Reviewer hmy9 (Part IV)**
>
> **W4&Q1: Unsupported and unclear methodology.**
>
> We appreciate the reviewer’s concerns and provide a detailed explanation below to clarify our methodology, specifically the transition from a bilevel to a single-level optimization and the rationale for our attack objective.
>
> **Transition to Single-Level Optimization**
>
> The original bilevel optimization involves an inner loop where the model adapts to test samples, including poisoned and benign samples, and an outer loop to optimize the attack objective. This structure is computationally intensive and impractical under the constraints of the RTTDP setting. To address this, we adopt the approximation strategy used in DIA [3] and make two key adjustments:
>
> 1. **Discarding the Inner Optimization**:
>
>     - In DIA [3], the inner optimization is approximated by assuming $\theta_t^* \approx \theta_t$, where $\theta_t^*$ represents the parameters after a full adaptation step, and $\theta_t$ represents the current parameters.
>     - This approximation is justified as TTA models typically update minimally during a single minibatch iteration, resulting in minor perturbations to $\theta_t$. Thus, the approximation retains practical relevance while simplifying the problem.
>
> 2. **Surrogate Model for Online Parameters**:
>
>     In the RTTDP protocol, direct access to online model parameters $\theta_t$ is unrealistic. Instead, we replace $\theta_t$ with the surrogate model parameters $\hat{\theta}_t$, which are accessible and trained to approximate the online model’s behavior.
>
> 3. **Final Optimization Objective**:
>
>     After these adjustments, the optimization simplifies to a single-level objective, as shown in Eq. 4 of the manuscript. This formulation allows efficient generation of poisoned samples while adhering to the realistic constraints of RTTDP.
>
> Additionally, we derive the **in-distribution attack objective**:
>
> Our proposed attack leverages the dependence of TTA models on self-training mechanisms, which aim to maximize confidence on pseudo-labels for adaptation. The core idea is as follows:
>
> 1. **Crafting Poisoned Samples**:
>     - The poisoned samples are constrained to maintain the shallow feature distribution of benign samples, satisfying $D(P_a, P_{ab})=0$ where $P_a$ and $P_{ab}$ denote the shallow feature distributions of poisoned and benign samples, respectively.
>     - However, these samples are intentionally manipulated to induce incorrect predictions by $\mathcal{L}_{atk}$. The model perceives these samples as valid but reinforces erroneous patterns during self-training.
>
> 3. **Reinforcing Erroneous Information**:
>     When the TTA model adapts to poisoned samples, it learns and reinforces incorrect associations. This creates a vulnerability, as future test samples with similar shallow feature distributions are more likely to be misclassified by the online model.
>
> 5. **Exploiting TTA Dependence on Test Data**:
>     Since TTA methods iteratively adapt using incoming test samples, our approach leverages this dependency to propagate the error induced by poisoned samples throughout the adaptation process.
>
> [3] Uncovering Adversarial Risks of Test-Time Adaptation
>
>
> **W6: Missing details on attack queries.**
>
> Under our RTTDP protocol, all methods generate the poisoned data based on the offline model (source model or surrogate model), and they all use the 40-iterations PGD attack as the optimized strategy (Please refer to the line 339 to 342 in the manuscript, different methods only change the optimized objective/loss). Each poisoned data queries the online model (TTA model) only for one time to obtain the predictions, like the normal users' samples.
>
> **MP1: The asymmetry KLD.**
>
> Thanks for your suggestions. We would clarify clearly about the used asymmetry KLD in the revised manuscript. We leverage the symmetry KLD loss in Eq. 1 which would provide a more robust and smooth aligning process between two distributions.
>
> **MP2&3&4: The improvement on Equations, Figures and Background introduction.**
>
> Thanks for your suggestions. We would improve the readability and conciseness of the equations, Figure 1 and the background introduction in our revised manuscript.
>
> ---
>
> Thank you for your valuable comments. We hope the above clarifications and additions comprehensively address your concerns. Your insights have been instrumental in improving the quality of our manuscript. We sincerely appreciate your support and constructive suggestions!
>
> Best regards,
>
> The Authors

---

> > ### Comment · Reviewer_hmy9 · 2024-11-26
> > **Response to the authors**
> >
> > **Distinction between RTTDP and TePA in Table 1**
> > The distinction between RTTDP and TePA remains unclear. What prevents explicitly TePA from being utilized in an online context? Both approaches target pre-trained models, so it seems feasible to run TePA by enabling attacks on the initial model architecture and weights, then evaluating the transferability of the attack.
> > I apologize if my initial comment was unclear, but could the authors clarify why TePA cannot be adapted to the same context as RTTDP? RTTDP is designed for online data, while TePA operates with offline data. However, since Table 2 compares the two approaches, the differences seem insufficient to justify this distinction.
> >
> > **Use of Projected Gradient Descent**
> > Minor. The paper refers to the optimization procedure as the PGD attack, however, to be precise, it seems the authors rely on a more general Projected Gradient Descent optimization approach to craft the poisoning data. PGD attack is just a specific implementation of Projected Gradient Descent optimization in practice for crafting adversarial examples.
> >
> > **Query usage and comparisons**
> > If all approaches use the same number of queries, the argument in lines 53–55 regarding query efficiency should be removed, as it does not represent a unique contribution. Furthermore, the choice of 40 queries appears arbitrary, and no ablation study assesses the impact of varying this parameter. What happens when the number of queries increases or decreases?
> >
> > **Asymmetry in KL Divergence**
> > The inclusion of two KLD terms in Equation 1 and the role of their asymmetry remain unclear and should be explicitly justified.
> >
> > **Overall clarity and presentation**
> > As noted by other reviewers, the paper needs more technical details, and its current presentation leaves several key questions unanswered. Clarity is a significant concern, as evidenced by the uniformly low Presentation scores (2 from all reviewers). While the authors have acknowledged these issues, no changes have been made during the rebuttal period, making it uncertain how the promised revisions and additional results will be incorporated into the final version.
> >
> > Given the above points, the paper requires significant revisions before its publication. Specifically, it needs:
> > 1. A more transparent distinction between RTTDP and TePA.
> > 2. Stronger experimental support, including ablation studies on parameters such as query counts.
> > 3. Better justification and explanation of methodological choices, including the optimization approach, KLD terms, and experimental setup.
> > 4. Improved presentation of the threat model and contributions relative to the state of the art.

---

> > > ### Author Response · Authors · 2024-11-27
> > > **Response to the Reviewer hmy9 (Part I)**
> > >
> > > We highly appreciate your professional and rigorous comments. These feedback gives us the opportunity that we are eager for to further improve this work. Regarding the revision manuscript, we are now trying our best to integrate all changes, including revised descriptions, additional discussions and additional experiments into the manuscript. Due to the time constraint, we aim to update the manuscript by 11:59pm Nov 27 AoE.
> > >
> > > **1. Distinction between RTTDP and TePA in Table 1**
> > >
> > > Thanks for the suggestion to further clarify the distinction between RTTDP and TePA. We summarize the distinction as follows and we believe the assumptions made by TePA make it fall under an offline method.
> > >
> > > - TePA employs a fixed surrogate model before test-time adaptation begins for generating poisoning which qualify the method as an offline method. The surrogate model is obtained by training a separate model (different architecture from the target model) using the same source dataset. For example, on TTA for CIFAR10-C, if the target model, i.e. the model deployed for inference and is subject to test-time adaptation, is ResNet18, TePA employs VGG-11 as the surrogate model and trains VGG-11 on the same source training dataset (CIFAR10 clean training set). This is evidenced from the source code released by official repository [A] and the descriptions in TePA "we assume that the adversary has background knowledge of the distribution of the target model’s training dataset. This knowledge allows the adversary to construct a surrogate model with a similar distribution dataset" [B].
> > > - TePA employs the fixed surrogate model to generate poisoned dataset $x^\prime$. Then generated poisoned dataset is fed to test-time adaptation to update model weights. **Afterwards**, TTA is further conducted on clean testing data for model update and performance evaluation. The above practice is evidenced by the source code [C]. **The segregation of data poisoning and TTA steps further support the claim that TePA should be classified as an offline approach**.
> > > - Finally, TePA could be adapted to online fashion and we made such an adaptation to TePA for comparison in Tab. 2-4 of the manuscript. Specifically, we use TePA to generate poisoning against the initial surrogate model and inject the generated poisoning into the testing data stream, i.e. placing poisoning in **between benign testing batches**. In this way, poisoning will affect TTA in an online fashion. We believe this is the most fair way to compare RTTDP with TePA.
> > >
> > >
> > > [A] https://github.com/tianshuocong/TePA/tree/main
> > >
> > > [B] Cong, Tianshuo, et al. "Test-time poisoning attacks against test-time adaptation models." 2024 IEEE Symposium on Security and Privacy (SP). IEEE, 2024.
> > >
> > > [C] https://github.com/tianshuocong/TePA/blob/main/TENT/poison_tent.py
> > >
> > >
> > > **2. Use of Projected Gradient Descent**
> > >
> > > We greatly appreciate the thoroughness of the feedback. The projected gradient descent (PGD) algorithm was originally designed to address constrained optimization problems [D]. In this context, the constraint serves to truncate the parameter update step, enabling efficient gradient-based optimization. To ensure a fair comparison among different data poisoning methods, we utilize the PGD algorithm to learn the poisoning (additive noise). We will clarify this point in the revised manuscript.
> > >
> > >
> > >
> > > [D] Boyd, Stephen, and Lieven Vandenberghe. Convex optimization. Cambridge university press, 2004.

---

> > > ### Author Response · Authors · 2024-11-27
> > > **Response to the Reviewer hmy9 (Part II)**
> > >
> > > **3. Query usage and comparisons**
> > >
> > > We appreciate the insightful comment. The concern about the number of queries mainly arises from the practice of the existing work, DIA, which assumes access to online (target) model for generating poisoning. DIA attempts to query online model 500 times [E], for PGD optimization, to generate a poisoned sample. Repeatedly querying the online model may alert the system.
> > > In general, the **number of allowed queries to the target model** should be taken into consideration . We do not claim limiting the number of queries as a unique technical contribution of this work, but instead we highlight this concern when investigating the adversarial risks of TTA models.
> > >
> > > Regarding varying query attempts, we add an additional evaluation as follows. Nonetheless, we want to highlight that the **query attempts do not have to be limited for our method because all queries are submitted to the surrogate model** rather than the target model. More queries simply makes generating poisoning slower. In this study, we vary the query steps from 10 to 60 for projected gradient descent optimization. The results in the table below suggest increasing the number of queries could improve the performance at the low query budget. When the budget is increased to beyong 40 queries, the performance saturates. We draw the conclusion that allowing sufficient queries to the surrogate model is necessary for generating effective data poisoning and, importantly, this procedure will not create alert to the target model.
> > >
> > > The results are obtained on the CIFAR10-C dataset with a Uniform attack frequency. We evaluate varying attack query steps for two TTA methods under their respective strongest attack objectives.
> > > | TTA Method | 10 | 20 | 30 | 40 | 50 | 60 |
> > > |-|:-:|:-:|:-:|:-:|:-:|:-:|
> > > |TENT (NHE Attack)|66.95|74.36|74.34|73.86|73.51|73.66|
> > > |EATA (BLE Attack)|35.73|39.70|42.36|45.20|45.99|45.89|
> > >
> > > [E] https://github.com/inspire-group/tta_risk/blob/main/conf.py#L150
> > >
> > > **4. Asymmetry in KL Divergence**
> > >
> > > Thanks for the question regarding the specific design. We adopt a common practice to symmetrize KL Divergence (KLD). We follow the definition made in the manuscript the **forward KLD** $KLD(h(x_i;\theta_t)||h(x_i;\hat{\theta}_t))$ and **reverse KLD** $KLD(h(x_i;\hat{\theta}_t)||h(x_i;\theta_t))$.
> > >
> > > Forward KLD emphasizes penalizing discrepancies where the distilled (surrogate) model $\hat{\theta}_t$ assigns low probability to samples that the source (real-time target) model $\theta$ deems important. It encourages the distilled model to mimic the behavior of the target model by focusing on areas of high confidence in $\theta$'s posterior.
> > >
> > > Reverse KLD, in contrast, focuses on matching $\theta$'s predictions where $\hat{\theta}$ assigns high probabilities. This can result in sharper, more focused distributions but might dismiss less probable regions of $\theta$
> > > posterior.
> > >
> > > The **symmetric KLD** balances the above two objectives. The forward KLD may be more suitable when surrogate model is significantly smaller than the target model and the objective is the allow the surrogate model to mimick the target model's certainty. When the surrogate model is of the same capacity with target model, using the symmetric KLD may better align the two models in both high confident and low confident predictions. In this work, the capacity of surrogate is similar to target model. Thus, we hypothesize that the symmetric KLD could be better.
> > >
> > > We further use empirical observations in the table below to support the hypothesis. With symmetric KLD the performance is slightly better than using the forward KLD.
> > >
> > > ||Symmetric KLD | $KLD(h(x_i;\theta _t)\|\|h(x_i;\hat{\theta}_t))$|
> > > |-|:-:|:-:|
> > > |TENT (NHE Attack)|73.86|**74.35**|
> > > |EATA (BLE Attack)|**45.20**|43.99|
> > > |SAR (BLE Attack)|**26.80**|26.35|
> > >
> > >
> > > Nevertheless, we do acknowledge that both symmetric KLD and forward KLD give competitive results. The choice depends on computation affordability and empirical observations.
> > >
> > > **5. Overall clarity and presentation**
> > >
> > > Finally, we highly appreciate the constructive comments given by all reviewers and managed to address the major comments as point-wise responses. Due to the time constraints, we posted the responses earlier than updating the manuscript. The updates will be reflected in the manuscript before the deadline. We are still eager to hear more comments and recommendations to improve this work. These constructive comments will help us further strengthen this work.
> > >
> > > ---
> > >
> > > Best regards,
> > >
> > > The Authors

---

> > > > ### Author Response · Authors · 2024-12-02
> > > > **Follow-up on the above Response**
> > > >
> > > > Dear Reviewer hmy9,
> > > >
> > > > Based on your valuable feedback, we have expanded our experiments to include two ablation studies, i.e., **Query Counts and Symmetric KLD**, and more detailed clarifications on the **distinction between RTTDP and TePA**. We have also **updated the revision** as you suggested. As we approach the end of the discussion period, we would be grateful if you could review our response and consider our revised manuscript. Your constructive comments have helped us to significantly improve our work, and we believe that we have thoroughly addressed your concerns.
> > > >
> > > > Thank you very much for your time and detailed review. We welcome any further additional questions or requests for clarification.
> > > >
> > > > Yours sincerely,
> > > >
> > > > The Authors

---

> > > > > ### Comment · Reviewer_hmy9 · 2024-12-02
> > > > >
> > > > > I would like to thank the authors for their huge work, which has undoubtedly enhanced the paper, and for this reason, I have increased my score. The rebuttal introduced many new experiments and highlighted a general need to improve the presentation of the contributions and results obtained. In fact, I believe the paper can still improve significantly in terms of presentation.

---

> > > > > > ### Author Response · Authors · 2024-12-03
> > > > > >
> > > > > > Thank you for your constructive comments that significantly improved our work. We will further improve our presentation for camera ready.

---

### Official Review · Reviewer_AP94 · 2024-11-04

**Soundness:** 3
**Presentation:** 2
**Contribution:** 2
**Rating:** 6
**Confidence:** 4

**Summary:**

The paper deals with data poisoning in the test time adaptation setting. The paper exposes some issues with existing poisoning attacks in this setting, and goes on to propose certain alterations/additions to existing poisoning objectives, and evaluates results on several datasets, with several test time adaptation methods.

**Strengths:**

* The experimentation is thorough, and generally well presented.
* The authors do a great job of pointing out issues with existing poisoning attacks against TTA training.
* The authors motivate and propose a poisoning scheme of their own, and show its effectiveness in several settings.

**Weaknesses:**

* The area of data poisoning in the TTA setting is a bit niche, and to be honest, doesn't seem to present many challenges beyond existing data poisoning settings. So called "availability" attacks have been introduced in [1,2], and several other works. These should at least be cited, and probably compared against as there's a decent amount of overlap.
* The real "value add" for the authors, in my view, is the addition of the feature clustering regularizer. The other contributions (BLE, notch loss, etc.) seem to be very slight modifications of existing poisoning attacks, or even just existing adversarial attacks.
* The related work shouldn't be at the end. It would be very helpful for readers to have some more info on TTA at the beginning of the work. But if you're going to keep it at the end, you NEED to cite things earlier, as it's totally unclear what things like TENT, RPL, etc. are when they're introduced. It seems like in section 4.2, only the acronyms are introduced, with no explanation, and no citation to click on and find more information. Note: I didn't deduct any "points" for this weakness, but it really needs to be addressed.

[1] Huang, Hanxun, et al. "Unlearnable examples: Making personal data unexploitable." arXiv preprint arXiv:2101.04898 (2021).

[2] Fowl, Liam, et al. "Adversarial examples make strong poisons." Advances in Neural Information Processing Systems 34 (2021): 30339-30351.

**Questions:**

* Does this only work against losses $\mathcal{L}_{tta}$ that are unsupervised? It would be nice to explain this a bit more and give some examples of what this loss function looks like in the main body.
* Do you ever specify the attack budget? I couldn't find it in the paper. In A.2, you define this quantity, $r$, but I don't ever see details for it. Is this the same thing as $b$ in Table 8?
* What are the "Source" numbers listed in Tables 2,3,4? Is this just the success of standard adversarial attacks?

---

> ### Author Response · Authors · 2024-11-24
> **Response to Reviewer AP94 (Part I)**
>
> **W1: Comparison between our proposed method and existing works [1,2].**
>
> Thank you for your suggestion. We will discuss and compare the two papers [1,2] in the revised manuscript. These papers focus on traditional data poisoning task, which differ significantly from the RTTDP setting we propose. Below, we outline the key differences in the goals, theoretical definitions, and corresponding methods between the traditional data poisoning and RTTDP tasks.
>
> **The Goal:**
>
> - **Traditional Data Poisoning (DP)** [1,2] aims to poison **all the training data** used to **train a model from scratch** under a **fully supervised protocol until convergence**, in order to perform poorly on normal test samples.
>
> - **Our proposed RTTDP** investigates the robustness of TTA methods against injecting some poisoned data into the test data stream. Therefore, it first follows the TTA pipeline where the online model is **initialized as a source pre-trained model**, and the TTA model uses each minibatch of test data to update the online model via **an unsupervised TTA loss**, and instantly produces predictions for the current test data. In RTTDP, the poisoned samples are injected into the TTA pipeline like normal test samples, and the performance is measured on benign users' (normal) samples.
>
> **Theoretical Definitions:**
>
> - In line with [1,2], we define **traditional data poisoning** as follows, where $\mathcal{T}_{te}$ and $\mathcal{T}\_{tr}$ denote the test and training sets, and $\mathcal{L}$ usually represents the cross-entropy loss:
>
>     $$\hat{\epsilon} = \arg\max_{\epsilon}\sum_{(x_i,y_i)\in\mathcal{T}_{te}} \mathcal{L}(f(x_i;\theta(\epsilon)),y_i)$$
>
>     $$\text{s.t.} \quad \theta(\epsilon) = \arg\min_\theta \sum_{(x_i,y_i)\in\mathcal{T}_{tr}} \mathcal{L}(f(x_i + \epsilon_i; \theta), y_i)$$
>
> - **Our RTTDP** follows a DIA optimization objective, where $\mathcal{B} _a$ and $\mathcal{B} _b$ represent poisoned and benign minibatch data, $\mathcal{L} _{atk}$ is the attack loss (e.g., maximizing cross-entropy), and $\mathcal{L} _{tta}$ is the unsupervised loss for TTA:
>
>     $$\mathcal{B} _a = \arg\min _{\mathcal{B} _a} \sum _{(x,y)\in\mathcal{B} _b} \mathcal{L} _{atk}(f(x;\theta _t^*(\mathcal{B} _a \cup \mathcal{B} _b)), y)$$
>
>     $$\text{s.t.} \quad \theta _t^*(\mathcal{B} _a \cup \mathcal{B} _b) = \arg\min _\theta \mathcal{L} _{tta}(f(\mathcal{B} _a \cup \mathcal{B} _b;\theta _t))$$
>
>     However, this formula presents challenges:
>     - **Inner optimization challenge**: The TTA loss $\mathcal{L} _{tta}$ and the online model $f(x;\theta)$ do not allow backward gradients to the poisoned samples $\mathcal{B} _a$. Moreover, black-box attack methods, which rely on repeated queries for gradient estimation, are impractical since each batch of poisoned samples requires numerous queries, and the online model is updated after each query.
>     - **Outer optimization challenge**: In RTTDP, we cannot observe benign user samples when generating poisoned samples. This is equivalent to not having access to $\mathcal{T}_{te}$ in the traditional data poisoning problem. Thus, directly targeting validation data would not effectively evaluate the risk of poisoned samples in the TTA model.
>
> - To address the inner optimization problem, we approximate $\theta _t^* \approx \theta _t$ since each minibatch updates the online model only slightly. The objective then becomes:
>
>     $$\mathcal{B} _a = \arg\min _{\mathcal{B} _a} \sum _{(x,y)\in\mathcal{B} _b} \mathcal{L} _{atk}(f(x;\theta _t(\mathcal{B} _a \cup \mathcal{B} _b)), y)$$
>
>     where $\theta(\mathcal{B} _a\cup\mathcal{B} _b)$ indicates forwarding $\mathcal{B} _a\cup\mathcal{B} _b$ to update the BN statistics. This objective would lead to a trivial solution that $\mathcal{B} _a$ is effective only for the current $\mathcal{B} _b$ data through easily introducing biased normalization in each BN layer, and it has little effect while $\mathcal{B} _a$ and $\mathcal{B} _b$ are in seperated batch. Furthermore, the adversary cannot observe $\mathcal{B} _b$ in practice.
>
> - To address the outer optimization problem, we modify the objective function using the PAC learning framework that constrains the shallow feature distributions of $\mathcal{B} _a$ and $\mathcal{B} _{ab}$ are similar, as shown in the manuscript (Eq. 2-4). The final objective becomes:
>
>     $$\mathcal{B} _a = \arg\min _{\mathcal{B} _a} \sum _{(x,y)\in\mathcal{B} _a} \mathcal{L} _{atk}(f(x;\hat{\theta} _t(\mathcal{B} _a)), y), \quad \text{s.t.} \ D(P _a, P _{ab}) = 0$$
>
>     where $\hat{\theta}_t$ is the surrogate model used due to the inaccessibility of the online model $\theta_t$.

---

> ### Author Response · Authors · 2024-11-24
> **Response to Reviewer AP94 (Part II)**
>
> **Corresponding Methods:**
>
> - **Traditional Data Poisoning**:
>     - [1] generates adversarial noise by minimizing the cross-entropy loss of training samples on a randomly initialized model. The poisoned samples produced with this noise prevent a randomly initialized model from learning true semantic information.
>     - [2] uses a pre-trained model to generate noise that confuses model predictions, i.e., $\hat{\epsilon} = \arg\min _{\epsilon} \sum _{(x _i,y _i)\in\mathcal{T}} \mathcal{L}(f(x _i + \epsilon _i; \theta^*), \hat{y _i})$, where $\hat{y _i} = y _i + 1$ and $\theta^*$ is the pre-trained model.
>     - Both [1] and [2] aim to prevent randomly initialized models from learning useful semantic information.
>
> - **Test-time Data Poisoning**:
>     - TePA generates poisoned samples by maximizing entropy on the source model, aiming to blur the classification boundaries of the online model.
>     - Our proposed method generates poisoned samples by constraining their feature distribution to closely match that of the samples before poisoning, causing the online model to learn incorrect information about the internal distribution.
>
>     Both approaches aim to make the source model forget the source domain knowledge and distort the classification boundary.
>
> Additionally, we will supplement experiments using methods from [1,2] to generate poisoned samples and inject them into the online model following the RTTDP protocol. The results for the CIFAR10-C dataset are shown below, and we will include results for CIFAR10/100 and ImageNet datasets in the camera-ready version.
>
> |Attack Objective|TENT|EATA|SAR|ROID|Avg|
> |-|-|-|-|-|-|
> |No Attack|19.72|18.03|18.94|16.37|18.27|
> |Unlearnable Examples[1]|32.61|20.11|19.23|17.80|22.44|
> |Adversarial Poisoning[2]|19.60|18.94|19.90|**19.12**|19.39|
> |BLE Attack(Ours)|54.07|**45.20**|**26.80**|19.06|36.28|
> |NHE Attack(Ours)|**73.86**|29.73|24.56|17.00|36.29|
> |Our Best|**73.86**|**45.20**|**26.80**|19.06|**41.23**|
>
> [1] Unlearnable examples: Making personal data unexploitable
>
> [2] Adversarial examples make strong poisons
>
>
> **W2: The Technical Contributions of Our Proposed Method.**
>
> We recognize that the main technical contribution of our method lies in the feature consistency regularization, which ensures that poisoned samples effectively attack the TTA model. We provide detailed derivations and explanations in the manuscript to support this.
>
> Regarding attack objectives, we categorize existing poisoning methods into two types: high-entropy and low-entropy attacks (as introduced in the manuscript). We then analyze the limitations of these objectives and improve them by proposing the BLE Attack and NHE Attack, respectively.
>
> **W3: Related Work.**
>
> Thank you for your suggestion. We will move the related work section to Chapter 2 to provide readers with a preview of the TTA methodology. The specific TTA methods are described in the "Benchmark TTA Methods" subsection of the Appendix due to space limitations in the main body.
>
> **Q1: Does this only work against losses $\mathcal{L} _{tta}$ that are unsupervised?**
>
> Yes, $\mathcal{L} _{tta}$ is an unsupervised loss used in TTA models. For example, TENT minimizes the entropy of test samples, i.e., $\mathcal{L} _{tta} = E _{x _i \in \mathcal{B} _t} \sum_k -p _{ik} \log p _{ik}$, where $p _i = f(x _i; \theta _t)$. EATA minimizes entropy while also including a Fisher regularization term to prevent forgetting source domain knowledge, i.e., $\mathcal{L} _{tta}=E _{x _i\in\mathcal{B} _t}Entropy(f(x;\theta _t)) + \beta R(\theta _t, \theta _0)$.
>
> **Q2: Do you specify the attack budget?**
>
> Yes, in the "Hyperparameters" subsection in the Appendix, we specify that the attack budget $r$ is 50% throughout the experiment. We also perform an ablation study on $r$ in Table 8. Additionally, we have corrected the confusing notation in A.3.2, where $r = 0.1, 0.2, 0.5$ was mistakenly written as $b = 0.1, 0.2, 0.5$ in the original manuscript.
>
> **Q3(A): What are the "Source" numbers listed in Tables 2, 3, and 4?**
>
> The "Source" numbers in these tables indicate the performance of the source pre-trained models tested directly on the test stream, without using TTA updates. Typically, TTA improves the performance of the pre-trained source model on the test data stream. However, when poisoned data is injected into the test stream, unsupervised TTA methods may harm the model's performance. More details about the TTA methods can be found in the "Benchmark TTA Methods" subsection of the Appendix.

---

> ### Author Response · Authors · 2024-11-24
> **Response to Reviewer AP94 (Part III)**
>
> **Q3(B): Is this just the success of standard adversarial attacks?**
>
> No, they are not the results of standard adversarial attacks. They are the results of the source pre-trained model directly validated on the test data stream as the traiditional testing. Because our RTTDP don't allow the adversary directly observe the benign samples that are used as the validation samples, so we don't have the results of standard adversarial attacks on benign samples.
> However, we can test the performance of standard adversarial attacks on our RTTDP setting by generating the poisoned samples using standard adversarial attacks and injecting them into the TTA pipeline to update the online model, and the results are still evaluated on the benign samples, as the results of MaxCE-PGD, GMSA-AVG, GMSA-MIN, AutoAttack in this rebuttal.
>
> ---
>
> Thank you for your valuable comments. We hope the above clarifications and additions comprehensively address your concerns. Your insights have been instrumental in improving the quality of our manuscript. We sincerely appreciate your support and constructive suggestions!
>
> Best regards,
>
> The Authors

---

> ### Author Response · Authors · 2024-12-02
> **Follow-up on Rebuttal Response**
>
> Dear Reviewer AP94,
>
> Based on your valuable feedback, we have supplemented our experiments to compare with two data poisoning methods, i.e., **Unlearnable Examples and Adversarial Poisoning**, and more detailed clarifications on **Comparison between our proposed method and existing works** to **highlight our challenges** and **Our technical contributions**. We have also **updated the revision** as you suggested. As we approach the end of the discussion period, we would be grateful if you could review our response and consider our revised manuscript. Your constructive comments have helped us to significantly improve our work, and we believe that we have thoroughly addressed your concerns.
>
> Thank you very much for your time and detailed review. We welcome any further additional questions or requests for clarification.
>
> Yours sincerely,
>
> The Authors

---

> ### Author Response · Authors · 2024-12-03
> **Follow-Up on Reviewer Feedback**
>
> Dear Reviewer AP94,
>
> With the discussion period ending in 10 hours, we would greatly appreciate hearing any further questions or concerns you may have. We are fully prepared to provide additional clarifications or responses as needed.
>
> To summarize, in our rebuttal, we have:
> - Conducted supplementary experiments comparing our method with two data poisoning approaches, **Unlearnable Examples** and **Adversarial Poisoning**.
> - Provided detailed clarifications on the **comparison between our proposed method and existing works**, highlighting the **challenges addressed** and our **technical contributions**.
> - **Revised the manuscript** in line with your valuable suggestions.
>
> We are pleased to note that our constructive discussions and revisions have resonated positively with other reviewers. For example:
> - Reviewer **8xpV** was fully convinced by our additional analysis on the limitations of query methods, details of proposed assumptions, and evaluation protocols.
> - Reviewer **hmy9** expressed satisfaction with the clarifications regarding online vs. offline protocols, additional evaluations against strong adversarial attacks, and our analysis of various design choices.
>
> We believe continued dialogue can further enhance the quality and clarity of this work. We look forward to your response and are happy to address any remaining questions.
>
> Thank you once again for your thoughtful feedback and support!
>
> Best regards,
> The Authors

---

> > ### Author Response · Authors · 2024-12-03
> > **Approaching the Deadline of Discussion Period**
> >
> > Dear Reviewer AP94,
> >
> > We kindly remind you that the discussion period will end in **2 hours**. May we ask if you feel that our responses above have adequately addressed your concerns? If so, we would sincerely appreciate if you could consider raising your rating accordingly.
> >
> > Thank you for your time and thoughtful review.
> >
> >
> > Best regards,
> >
> > The Authors

---

### Official Review · Reviewer_f4Wt · 2024-11-08

**Soundness:** 3
**Presentation:** 2
**Contribution:** 3
**Rating:** 6
**Confidence:** 3

**Summary:**

This paper examines the adversarial risks in test-time adaptation (TTA), highlighting that TTA’s exposure to test-time data poisoning may be less severe under realistic attack assumptions. The authors propose two TTA-aware attack objectives and a new in-distribution poisoning method that operates without access to benign data, revealing that TTA methods show greater robustness than expected and identifying defense strategies to enhance TTA’s adversarial resilience.

**Strengths:**

1. The paper is overall well-written with clear math definitions and extensive experiments and studies an important problem of data poisoning under weaker/realistic assumptions.
2. The paper studies the popular test-time attack problem from a novel perspective of weaker threat model.

**Weaknesses:**

1. Line 680, citation format wrong. Missing conference name.
2. Figure 1 is helpful but visually confusing and ovewhelming. Please explain what B_a, B_ab, B_t, B_b in the figure description section. A term (-\lambda L_{reg} and L_{atk}) is confusing to put with the full equation . Also, it is better to present the attack objective in separate part than to fit it in Figure 1.
3. It will be more helpful to have a graph that assign the current popular attack methods into different buckets, where each bucket has different threat model.

**Questions:**

1. Has the author investigate the effect of the ratio of between the adversarial example and the benign examples?
2. Did the author experiment and compare with harder attack like in [1] that requires access to benign examples. Or is there quantitative measurement on the tradeoff when relaxing to realistic attacks.
3. Is it possible to derive any formal guarantee on the attack effectiveness when relaxing the different constraints?



[1]. Chen, J., Wu, X., Guo, Y., Liang, Y., and Jha, S. Towards evaluating the robustness of neural networks learned by transduction. In Int. Conf. Learn. Represent., 2022

---

> ### Author Response · Authors · 2024-11-24
> **Response to Reviewer f4Wt**
>
> **W1: Citation format error**
>
> Thank you for pointing this out. We have corrected the citation format in the revised manuscript.
>
> **W2: Clarification for Figure 1**
>
> We appreciate your valuable suggestions. To address the confusion, we will expand the descriptions of the notations in the caption of Figure 1. Specifically:
> - $\mathcal{B} _{ab}$ and $\mathcal{B} _a$ denote the adversary's samples, where $\mathcal{B} _{ab}$ represents the benign samples prior to poisoning, and $\mathcal{B} _a$ refers to the poisoned samples.
> - $\mathcal{B} _b$ represents normal/benign user samples.
> - $\mathcal{B} _t$ denotes the input data to the TTA pipeline at timestamp $t$.
>
> As suggested, we will further refine Figure 1 in the revised manuscript for improved clarity.
>
> **W3: Graphical classification of threat models**
>
> Thank you for the suggestion. We will include a graph classifying the threat models used in various attack methods. This addition will be provided in the Appendix of the revised manuscript.
>
> **Q1: Impact of the ratio between benign and poisoned samples**
>
> We have analyzed the impact of varying poisoned sample ratios in Table 8 of the Appendix. The results demonstrate a reasonable trend: the error rate of benign samples predicted by the online model improves gradually as the proportion of poisoned samples increases.
>
> **Q2: Comparison with harder attacks requiring access to benign examples**
>
> Thank you for recommending valuable related work[1]. We will incorporate a discussion of this method in our revised manuscript. Specifically, we have implemented the GMSA attacks[1], including GMSA-MIN and GMSA-AVG, in the context of our RTTDP protocol. In this setup, poisoned samples were generated using the GMSA attack methods and subsequently injected into the TTA model. The attack success rate was then evaluated on other benign user samples. The results on CIFAR10-C dataset with `Uniform` attack frequency are shown in the below table.
>
> |Attack Objective|TENT|EATA|SAR|ROID|Avg|
> |-|-|-|-|-|-|
> |No Attack|19.72|18.03|18.94|16.37|18.27|
> |GMSA-MIN[1]|35.92|22.78|19.99|18.65|24.33|
> |GMSA-AVG[1]|38.80|21.89|19.95|18.51|24.79|
> |BLE Attack(Ours)|54.07|**45.20**|**26.80**|**19.06**|36.28|
> |NHE Attack(Ours)|**73.86**|29.73|24.56|17.00|36.29|
> |Our Best|**73.86**|**45.20**|**26.80**|**19.06**|**41.23**|
>
>
> [1] Towards evaluating the robustness of neural networks learned by transduction.
>
>
> **Q2&3: Is it possible to derive any formal guarantee on the attack effectiveness when relaxing the different constraints?**
>
> In our ablation study, as presented in Table 5, we analyzed scenarios involving access to online model weights and the impact of excluding the feature consistency constraint. Our observations are as follows:
>
> 1. **Impact of Feature Consistency Constraint**
>
>     - In the absence of a feature consistency constraint, low-entropy attacks typically generate poisoned samples that fail to effectively influence benign samples through TTA updates.
>     - Conversely, high-entropy attacks, such as those that maximize entropy or utilize NHE, prove to be more effective. This is because the TTA gradients computed from poisoned samples with high-entropy predictions are significantly larger, causing substantial perturbations to the source model during updates.
>
> 2. **Effectiveness of Feature Consistency Constraint**
>
>     The inclusion of our proposed feature consistency constraint greatly enhances the effectiveness of the attack, regardless of whether poisoned samples are generated using the surrogate model or the online model.
>
> 3. **Access to Online Model Weights**
>
>    The average attack success rate across all TTA methods is highest when the attacker has access to the online model, enabling the generation of more accurate poisoned samples.
>
> On the other hand, assuming access to benign user samples presents its own challenges. If such access were feasible, an adversary could modify benign samples directly through adversarial attacks before they are injected into the online model, rather than introducing poisoned samples.
>
>
> ---
>
> Thank you for your valuable comments. We hope the above clarifications and additions comprehensively address your concerns. Your insights have been instrumental in improving the quality of our manuscript. We sincerely appreciate your support and constructive suggestions!
>
> Best regards,
>
> The Authors

---

> > ### Comment · Reviewer_f4Wt · 2024-11-27
> >
> > The authors have addressed all my questions. I will keep my current rating.

---

> > > ### Author Response · Authors · 2024-11-28
> > >
> > > We thank you again for your constructive comments and positive feedback!

---

### Official Review · Reviewer_8xpV · 2024-11-09

**Soundness:** 3
**Presentation:** 3
**Contribution:** 3
**Rating:** 8
**Confidence:** 4

**Summary:**

This paper investigates adversarial risks in test-time adaptation (TTA), which updates model weights during inference to counter distribution shifts. The authors argue that the assumed threat model of prior work is unrealistic and gives the attacker too much access. They propose a more restricted attack model called Realistic Test-Time Data Poisoning (RTTDP), assuming grey-box access to the model and no access to the benign samples of other users. The authors propose a new attack method outperforming previous work in this new setting.

**Strengths:**

**More Realistic Threat-Model**

The paper critically examines the assumptions made in prior work regarding the threat model of poisoning attacks against TTA systems, particularly regarding the adversary’s capabilities. In particular, the relaxation of (i) white-box access to gray-box access and (ii) restricting access to the targeted samples makes the setting much more realistic and, therefore, the findings more significant and relevant.

**A New Attack Rooted in Theoretical Insights**

The proposed method cleverly extends on previous attacks, making it effective in the new, more restricted attack setting. The method is rooted in theoretical insights that nicely explain the various parts and modifications. It demonstrates that TTA is still vulnerable to poisoning attacks even with more restrictive assumptions.

**Thorough Evaluation**

The experimental evaluation is thorough. It spans three different datasets with increasing complexity (CIFAR10-C, CIFAR1-C, ImageNet-C), multiple different TTA methods, and compares against two different baselines. The evaluation also includes an ablation study, as well as first look at potential defenses, which I appreciate. The defenses show promise but are unable to fully recover the functions of the original models.

**Weaknesses:**

**Improved Assumptions Still Strong**

While the gray-box assumption is a significant improvement over the white-box assumption in some prior work, it still assumes the adversary's access to the original model (before TTA) and data from the same distribution as the victim benign user. Both of these assumptions are fairly strong and could ideally be relaxed further to, e.g., black-box access.

**Unclear Details**

Several important details of the method remain unclear and should be included in the paper. The most important ones include: why repeated querying of the model is forbidden (L181), how the surrogate model can be trained without querying (Section 3.1), why we can assume to know $\mathcal{B}_{a, t-1}$, and how “Uniform” and “Non-Uniform” attacks are executed. Please refer to “Questions” for more details and additional comments.

**The Presentation Can Be Improved**

The method’s presentation could be improved in multiple ways:

- Figure 1 gives a good overview of the method. It could benefit from some simplifications to reduce visual clutter; see “Suggestions for Improvements” for detailed suggestions.
- Figure 2 is very small and should be enlarged. The four subplots are also unrelated and referred to from different parts of the paper. I suggest splitting it into individual figures and moving those to the corresponding sections.
- In general, the paper’s English is easy to read. However, some parts are hard to understand due to grammatically wrong or overly complex sentences (e.g. L431, L259, …). The paper would benefit significantly from a careful revision, possibly with an English language tool.

**No Code Available**

No code was available for review, and the authors did not specify whether it will be released upon publication.

**Questions:**

**Questions**

L181: Why is repeated querying prohibited? I understand that the model is constantly updated throughout the querying, but is this really a problem?

Section 3.1: How exactly is the model distilled? L181 says repeated querying is prohibited, but my understanding is that training a surrogate model requires precisely such querying? How can you query for a single $\theta_t$ if it changes by the very fact that it is being queried?

L232/233: How can we assume to know $\mathcal{B}_{a, t-1}$? Other users could have queried the model an unknown number of times between training the surrogate and launching the attack.

Table 2-4: What do “Uniform” and “Non-Uniform” Attack Freq. refer to? I could not find this mentioned anywhere in the paper. The appendix has a figure, but also no sufficient explanation. Since this is a central part of the main results tables, it should be carefully explained in the main part of the paper.

Table 2-4: What does “Source” refer to? this should also be explained.

L259/260: I don’t understand what this sentence means. What is “the optimized”, what is the “optimizing objective”, what is the “problem”?

L431: I do not understand this sentence. Can you please rephrase?

L149/150: “the attacker is assumed to have no access to real-time model weights.” By whom is this assumed?

Table 1: what does “Offline” attack order refer to?

**Suggestions for Improvement**

Figure 1: The overview figure 1 gives over the method is great! I appreciate the difficulty in illustrating the complex system with many aspects. I would like to make some suggestions to simplify the figure, as it took me a long time to work through it, and I believe it would be much more helpful with slightly less detail that distracts from the core parts:

- I suggest entirely removing the two boxes “TTA Model” and “Attack Objectives”. They don’t add much information over the text, and it would allow the viewer to concentrate on the three parties in the system: the adversary, the benign user, and the TTA server.
- Removing the symbols for the attacker and user would further reduce visual clutter, and they are redundant to the titles of the boxes.
- I would stylize the images from the distributions - e.g. squares of one color per distribution - instead of using actual images from the dataset. This has several advantages: (i) it reduces visual clutter, (ii) it makes it immediately apparent which distribution they belong to, (iii) the “poison” symbol becomes more apparent (it took me a while to see the little devils).

There may be more opportunities for improvements, e.g., thicker lines for arrows, but I believe the three points above will already make the figure significantly easier to understand.

Figure 2: Please consider breaking these plots into individual figures.

- They are currently way too small, to the point where labels and points are barely readable when printed. At least doubling their size would be required.
- The several subplots are unrelated and even referenced from different sections of the paper. It would be much more helpful to have the figure close to the text that it refers to.

L298: You refer to Fig 2(c), which contains many acronyms that have not been introduced. E.g., TENT, EATA, SAR, NHE, BLE, …

Tables 5-7: These tables would be easier to read without the grey shading.

---

> ### Author Response · Authors · 2024-11-24
> **Response to Reviewer 8xpV (Part I)**
>
> **W1: Improved Assumptions Still Strong.**
>
> Thank you for your insightful comments. In this work, we relax two critical assumptions commonly made in prior research: (i) the poisoned samples are generated using a white-box (online) model, (ii) the poisoned samples are created by maximizing the error rate of benign users' (validation) samples. These assumptions represent significant limitations to the practical applicability of earlier studies.
>
> In response to the points raised by the reviewer:
>
> - We believe that obtaining the source model is relatively straightforward, as it is often a well-known model, such as an open-source foundation model or a pre-trained model derived from large-scale datasets. Additionally, an approximate source model can be distilled using a set of test data, further reducing the dependence on this assumption.
>
> - It is more practical to obtain the distribution of benign users' data than to access their specific samples. For instance, an adversary could target data from a specific environmental condition (e.g., rainy or foggy settings) where the data distribution can be reasonably approximated.
>
> We acknowledge the importance of exploring even more relaxed assumptions to enhance the practicality of our approach, and this will be a priority for future work. Nonetheless, we believe that our current study represents a significant advancement over existing methods, both in practicality and theory.
>
>
> **W2: Unclear Details.**
>
> We appreciate the reviewers’ careful reading of our manuscript and their valuable questions. Below are our detailed responses to each query:
>
> **Q1: L181: Why is repeated querying prohibited?**
>
> In traditional black-box attacks, adversarial gradients are estimated via repeated querying. However, obtaining a batch of poisoned samples through such methods typically requires thousands of queries. This approach is impractical in real-world online TTA scenarios because (i) excessive querying is easily detectable using straightforward monitoring strategies, and (ii) the TTA model updates with each query, rendering gradient estimates unreliable. To clarify clearly, we will revise the term "prohibited" to "unavailable" in the manuscript.
>
> **Q2: How exactly is the model distilled?**
>
> As shown in Fig. 2(a), the distilled surrogate model suffices for generating poisoned samples, achieving performance comparable to the online model. In our method, the surrogate model is updated iteratively (10 iterations using Eq. 1) based on the feedback from the last batch of injected poisoned samples. Importantly, these updates use the fixed feedback, eliminating the need for repeated queries to the online model.
>
> **Q3: How can we assume to know $\mathcal{B}_{a,t-1}$?**
>
> The notation $\mathcal{B}_{a,t-1}$ refers to poisoned data injected in the previous round. Between two launching attacks, an unknown number of benign user samples enter the TTA model. The surrogate model need not align perfectly with the real-time model parameters $\theta_t$; instead, it is sufficient for the surrogate model to approximate the lagged parameters $\hat{\theta} _t \approx \theta _{t-\delta}$, where $\delta$ indicates the timestamp gap between two batches of injected poisoned data. We will clarify this in the revised manuscript.
>
> **Q4: What do `Uniform` and `Non-Uniform` Attack Frequencies refer to?**
>
> These terms are defined in the “Evaluation Protocol” subsection of the Appendix due to space limitations in the main text. `Uniform` refers to poisoned samples being uniformly injected throughout the TTA pipeline, while `Non-Uniform` refers to concentrated injections at the beginning of each test domain.
>
> **Q5: What does "Source" refer to?**
>
> "Source" in Tables 2–4 denotes the baseline inference performance of the source pre-trained model without test-time adaptation. Details about TTA methods are provided in the “Benchmark TTA Methods” subsection of the Appendix.
>
> **Q6: The meaning of L259–L260.**
>
> Lines 248-257 discuss the inefficiency of generating poisoned samples using Eq. 3, as $\mathcal{B} _a$ only harms the TTA model when jointly injected with $\mathcal{B} _{ab}$, but it would waste half of query budget. The mismatch in feature distributions between $\mathcal{B} _a$ and $\mathcal{B} _{ab}$ causes normalization statistics in batch normalization (BN) layers to differ when $\mathcal{B} _a$ is forwarded alone versus with $\mathcal{B} _{ab}$. To address this, in lines 259-260, we introduce an additional constraint, $D(P _a, P _{ab}) = 0$, combining the optimizating objective $\mathcal{B}_a$ and optimized objective $\mathcal{B} _{ab}$ into a single objective. We will clarify this notation in the revised manuscript.
>
> **Q7: The meaning of L431.**
>
> Line 431 states that our proposed surrogate model outperforms the source model in generating poisoned data for PGD attacks and even slightly exceeds the online TTA model in Table 5. We will rephrase this sentence for clarity.

---

> ### Author Response · Authors · 2024-11-24
> **Response to Reviewer 8xpV (Part II)**
>
> **Q8: The meaning of L149/150.**
>
> In realistic scenarios, the attacker is unable to access the real-time model weights. The `assumed` used into this sentence is meant to correspond to the reason "The update is carried out on the cloud side" stated before. We will clarify clearly in the revised manuscript.
>
> **Q9: What does "Offline" attack order refer to?**
>
> The TePA protocol is classified as “Offline” because all poisoned samples are injected into the TTA model before any benign user samples are processed. In contrast, our RTTDP protocol injects poisoned samples into the TTA model alongside benign samples throughout the test stream, enabling online updates using mixed data. Thus, RTTDP is classified as an “Online” attack.
>
> We hope these responses address the reviewers' questions comprehensively.
>
> **W3: No Code Available.**
>
> We promise to release our all code, including the implementations of our proposed method and all competing methods, to provide more implemented details as soon as this manuscript is accepted.
>
> **For the improvement suggestions.**
>
> We thank the reviewer for the efforts to help us improve our manuscript representation. We would update the Figure 1 and Tables 5-7 according to the suggestions in the revised version. Due to the limited space, we could not break Figure 2 into individual figures.
>
> ---
>
> Thank you for your valuable comments. We hope the above clarifications and additions comprehensively address your concerns. Your insights have been instrumental in improving the quality of our manuscript. We sincerely appreciate your support and constructive suggestions!
>
> Best regards,
>
> The Authors

---

> > ### Comment · Reviewer_8xpV · 2024-11-29
> >
> > Thank you for your thorough response! It answers some of my questions. However, some points remain open:
> >
> > **Q1: L181: Why is repeated querying prohibited?**
> >
> > Changing prohibited to unavailable is nor really helping my understanding here. The question I have is (1) you say a model, which is available as a black-box to the attacker, cannot be queried. This is already an inconsistency. (2) You then query said model to train the surrogate. This does not make sense to me. I also believe that the claim that this is "easily detectable using straightforward monitoring strategies" is not obvious and needs some evidence.
> >
> > **Q3: How can we assume to know ?**
> >
> > Thank you for clarifying. How large to you set $delta$ in practice for your experiments? How different is the distribution of the "other" queries to yours? I would expect these two parameters to have an impact on the experiments.
> >
> > **Q4: What do Uniform and Non-Uniform Attack Frequencies refer to?**
> >
> > I did see that appendix. However, (1) the main result tables should be understandable without referring to the appendix. (2) the details of the evaluation protocol remain unclear to me. What is the ratio between benign user queries and attacks? How do the data distributions differ? What impact does this have on the results?

---

> > > ### Author Response · Authors · 2024-11-29
> > > **Response to Reviewer 8xpV (Part I)**
> > >
> > > We thank you for your valuable comments and constructive suggestions, which help us to further improve our work. Below are the answers to the above questions, which we hope will address your concerns.
> > >
> > > **Q1: L181: Why is repeated querying prohibited?**
> > >
> > > We would like to clarify that we never stated the model could not be queried. Instead, the TTA model can be queried just like any typically deployed test model. Specifically, when a batch of samples is fed to the TTA (online) model, the predictions for that batch are obtained, and simultaneously, the online model is updated by the TTA method.
> > >
> > > **Response to the First Question:** In Lines 221-223 of the manuscript, we originally wrote: *"However, in the test-time adaptation setting, the online model is continuously updated with each query during the inference phase. Thus, repetitive querying the model for gradient approximation or fitness evaluation is unavailable."* This statement highlights that, since the online model would update when a batch of test samples (whether poisoned or benign) query, the typical approach used in black-box attacks, **repetitive querying** for gradient estimation, is not feasible. **Traditional black-box attack** methods rely on querying a **static model** for gradient approximation, **but** this assumption does not hold for **TTA models**, which are **dynamic**. Moreover, querying the online model excessively (e.g., thousands of times to craft a single batch of poisoned samples) would be computationally inefficient and easily detectable [A].
> > >
> > > **Response to the Second Question:** The surrogate model is updated (distilled) after receiving predictions of the injected poisoned samples from the online model. To avoid repetitive querying, we keep the predictions obtained from the online model fixed and use Eq. 1 in our manuscript to distill the surrogate model for 10 iterations. This design prevents excessive interactions with the online model while maintaining the quality of the surrogate model updates.
> > >
> > > **Regarding Query Detectability:** In our rebuttal response, we noted that "excessive querying is easily detectable using straightforward monitoring strategies." This observation is supported by prior works [B, C], which emphasize that frequent queries to a black-box model can raise suspicion, prompting the development of query-efficient methods to mitigate this risk. Additionally, [D] proposed a specific strategy to detect query-based black-box attacks.
> > >
> > > **Summary of Our Perspective:** We argue that relying on repetitive querying of the online model, particularly for hundreds or thousands of iterations to craft poisoned samples, is impractical in realistic scenarios for two main reasons:
> > >
> > > 1. **Dynamic Updates of the Online Model**: The TTA model is not static and updates itself during the querying process, making repetitive querying for gradient estimation not feasible.
> > > 2. **Detection and Efficiency**: Excessive querying to generate poisoned samples is both inefficient and easily detectable.
> > >
> > > [A] Zhen Yu, et al. Query-Efficient Textual Adversarial Example Generation for Black-Box Attacks.
> > >
> > > [B] Tao Xiang, et al. Towards Query-Efficient Black-Box Attacks: A Universal Dual Transferability-Based Framework
> > >
> > > [C] Andrew Ilyas, et al. Black-box Adversarial Attacks with Limited Queries and Information.
> > >
> > > [D] Huiying Li, et al. Blacklight: Scalable Defense for Neural Networks against Query-Based Black-Box Attacks
> > >
> > >
> > > **Q3: How can we assume to know?**
> > >
> > > In line 1038, we define the notation $r$ as the attack budget which represents the ratio of the benign samples and poisoned samples. In most of experiments, we evaluate the attack performance with $r=50\%$, i.e. $\delta=2$, where $\mathcal{B}_a$ and $\mathcal{B}_b$ are interleaved under the `Uniform` attack order. Additionally, we have conducted the ablation study on $r$ as shown in Tab. 12 in Sec A.4.4.
> > >
> > > **Q: How different is the distribution of the "other" queries to yours?**
> > >
> > > Let me clarify the context to ensure we're aligned. Are you asking whether "other queries" refer to the queries from benign users' data $\mathcal{B}_b$? If so, here’s the explanation:
> > >
> > > In our work, we introduce a feature consistency regularization term to constrain the generation of poisoned samples. And as shown in Figure 2(b) of the manuscript, the distribution of our poisoned samples, optimized with the combined objective $\mathcal{L}{atk} + \lambda \mathcal{L}{reg}$, significantly overlaps with the distribution of benign samples.
> > >
> > > **Q4: What do Uniform and Non-Uniform Attack Frequencies refer to?**
> > >
> > > Thanks for your constructive suggestion. We will supplement the description of 'Uniform' and 'Non-Uniform' in the caption of the tables in the camera ready version.

---

> > > ### Author Response · Authors · 2024-11-29
> > > **Response to Reviewer 8xpV (Part II)**
> > >
> > > **How do the data distributions differ?**
> > >
> > > In our test data stream, the data distribution changes gradually over time, according to the continuous test time adaptation setting [E]. On whether CIFAR10-C, CIFAR100-C or ImageNet-C, there are 15 types of corrupted images, including gaussian noise, shot noise, impulse noise, defocus blur, glass blur, motion blur, zoom blur, snow, frost, fog, brightness, contrast, elastic transform, pixelate, jpeg compression. The differences among the corruptions can refer to the Fig. 2 in [F].
> > > Following [E], our test data stream includes all kinds of corrupted images and appears in chronological order, data distribution by data distribution, i.e. gaussian noise $\rightarrow$ shot noise $\rightarrow$ impulse noise $\rightarrow$ defocus blur $\rightarrow$ glass blur $\rightarrow$ motion blur $\rightarrow$ zoom blur $\rightarrow$ snow $\rightarrow$ frost $\rightarrow$ fog $\rightarrow$ brightness $\rightarrow$ contrast $\rightarrow$ elastic transform $\rightarrow$ pixelate $\rightarrow$ jpeg compression.
> > > We will add more details about the construction of test data stream in the camera ready version.
> > >
> > > [E] Qin Wang, et al. Continual Test-Time Domain Adaptation.
> > >
> > > [F] Yongyi Su, et al. Revisiting Realistic Test-Time Training: Sequential Inference and Adaptation by Anchored Clustering Regularized Self-Training.
> > >
> > > ---
> > >
> > > Best regards
> > >
> > > The Authors

---

> > > > ### Comment · Reviewer_8xpV · 2024-12-01
> > > >
> > > > Thank you for the additional clarifications.
> > > >
> > > > **Q1: L181: Why is repeated querying prohibited?**
> > > >
> > > > Thank you - I understand your point now. It may be helpful to change the phrasing a bit to avoid future readers running into the same issue - ideally expanding the discussion slightly as in the answers provided above. Including those references will also significantly strengthen your point.
> > > >
> > > > **Q3: How can we assume to know?**
> > > >
> > > > So to clarify - the assumption is that all users of the model have data from the same distribution - and this distribution changes over time.
> > > >
> > > > Regarding the evaluation protocol details (e.g., $r$): I believe many of my questions sem from the fact that almost all properties of the evaluation are deferred to the appendix. This makes understanding and interpreting the experiments challenging. I suggest moving the evaluation protocol section to the main paper, as this is crucial information. If space is limited, you could move some of the ablation experiments, or Section 5.5. to the appendix instead.

---

> > > > > ### Author Response · Authors · 2024-12-02
> > > > >
> > > > > Thank you for your comments! We will do our best to further improve the clarifications in the revised version. We will include an explanation of the evaluation protocol in the main text so that the reader is clearer about what we are doing.
> > > > >
> > > > > We are happy to discuss with the reviewer to consistently improve our manuscript!
> > > > >
> > > > > Thank you very much!
> > > > >
> > > > > The authors

---

> > > > > > ### Comment · Reviewer_8xpV · 2024-12-02
> > > > > >
> > > > > > Thank you for the constructive discussion. The revisions and additional results during the rebuttal have significantly improved the paper. I trust that the authors solve the remaining issues and further improve on the presentation for the camera ready. I increased my score accordingly.

---

> > > > > > > ### Author Response · Authors · 2024-12-03
> > > > > > >
> > > > > > > Thank you for your full support of our work. We promise to solve the remaining issues and further improve the presentation for camera ready. And we will release the implementation code of our work as soon as this work is accepted.

---

### Author Response · Authors · 2024-11-28
**Global Response**

Dear reviewers,

We highly appreciate the constructive and professional comments provided by the reviewers, which has significantly improve our manuscript in every way. The revisions made are marked with $\color{blue}{\text{``blue"}}$ in the revised paper. Below, we outline the specific revisions:

1. Visual clutter in Fig. 1. According to the reviewers' suggestions, we **simplified Fig. 1**, removing two panels with little information and some useless symbols, and improving some details. Also, we add more descriptions of the notations used into Fig. 1 in the caption.

2. **The related work**. To facilitate non-expert readers understand the background of our work, we have placed the relevant work in **Section 2** and supplement some advanced related work, e.g. **unlearnable examples, adversarial poisoning, GMSA and some surveys** suggested by the reviewers.

3. Some unclear details. In response to the **reviewers' questions**, we have revised unclear descriptions and corrected the corresponding sentences. These updates can be found in **Lines 190–192, 223, 260, 299–300, 337–341, 380, and 478–480**.

4. Comparison with **two data poisoning methods**. We have supplemented the experiments of two data poisoning method **"Unlearnable Examples" and "Adversarial Poisoning" on CIFAR10-C in Tab. 2**. Due to the time constraint, the subsequent experiments on CIFAR100-C and ImageNet-C will be added in the camera ready version.

5. **Distinction between RTTDP and TePA**. In **Appendix A.1.4**, we provide a detailed discussion to explain why we categorize TePA as an offline attack order and clarify how we adapt TePA into our realistic RTTDP protocol.

6. **Transition from Bilevel to Single-Level Optimization**. In **Appendix A.1.5**, we provide a detailed derivation and explanation of the process for transforming the original bilevel optimization problem into the final single-level optimization formulation. Additionally, we analyze the rationale behind the proposed formulation and its effectiveness from a TTA perspective.

7. **Experimental setups of DIA, TePA and RTTDP**. In **Appendix A.2**, we supplement the commonalities and differences among these relevant protocol in details, and clarify how we adapt DIA and TePA into our RTTDP for a fair comparison. It is worth emphasising that, all results in Tab. 2-4 are evaluated under our RTTDP protocol, which is an online attack order and is not allowed to access the online model weights and other users' benign samples. To minimise misunderstandings, we have added asterisks to DIA and TePA in **Table 2-4** to indicate that we have made the corresponding adaptations to differentiate the experimental setup from that of the original paper.

8. **Implementation Details and Reproducibility.** In **Section 5.1** and **Appendix A.3**, we provide additional implementation details for both the competing poisoning methods and our proposed methods. These details include the use of a **40-step Projected Gradient Descent (PGD) optimization algorithm** to optimize the objectives, the **threat model**, the **formulas for specific objectives**, and clarifications on the notations. We believe these supplementary details will help readers gain a deeper understanding of our experimental setup and facilitate reproducibility.


9. **Comparison with Advanced Adversarial Attack Methods**. In **Appendix 4.1**, we present experiments comparing our proposed poisoning methods with several advanced adversarial attack methods, such as **AutoAttack, GMSA-AVG, and GMSA-MIN**. We aim to investigate whether the adversarial effects of poisoned samples generated by these advanced methods can effectively transfer to benign users' samples within the RTTDP protocol. The results indicate that this is not easy.

10. **Ablation study on Query Count**. In **Appendix 4.2**, we conduct the ablation study on the query count used into the PGD algorithm to generate poisoned samples.

11. **Analysis on Symmetric KLD** used for surrogate model distillation. In **Appendix 4.3**, we compare the symmetric KLD and forward KLD used to distill the surrogate model.

---

Finally, we highly appreciate the constructive comments given by all reviewers again. We are still eager to hear more comments and recommendations to improve this work. These constructive comments will help us further strengthen this work.


Best regards

The Authors

---

### Author Response · Authors · 2024-12-01
**Willing to Address Additional Comments**

Dear reviewers,

We would like to highly appreciate all reviewers' efforts and time again in providing valuable comments and constructive suggestions for improvement of our submission. We hope that the clarifications and additional evaluations provided in the responses have addressed all reviewers' questions and concerns.

We are always ready to provide additional clarifications should you have any questions and concerns during the discussion period, due on 2nd Dec.

Thank you very much!

The Authors

---

### Meta-Review · Area_Chair_RHnG · 2024-12-23

**Metareview:**

"On the Adversarial Risk of Test Time Adaptation: An Investigation into Realistic Test-Time Data Poisoning" re-investigates assumptions made in previous work on data poisoning during test time augmentation, propose a new gray-box setting and provide a careful investgation in this, more realistic, setting.

Reviewers generally like this reinvestigation of attacks in the TTA setting, and the careful investigation in the submission. A number of concerns remain, such as the, still relatively large, attack budget necessary in this setting, compared to standard data poisoning, and the clarity of presentation of the results and the comparison to adversarial attacks. I do think it is in the authors' best interest to further improve the clarity of their presentation until the conference.

**Additional Comments On Reviewer Discussion:**

The authors extended their comparison and treatment of both classical data poisoning attacks and of classical adversarial attacks, based on reviewer feedback.

---

### Decision · Program_Chairs · 2025-01-22

Accept (Poster)